# VeriFlow: Modeling Distributions for Neural Network Verification

## Abstract

Formal verification has emerged as a promising method to ensure the safety and reliability of neural networks. Naively verifying a safety property amounts to ensuring the safety of a neural network for the whole input space irrespective of any training or test set. However, this also implies that the safety of the neural network is checked even for inputs that do not occur in the real-world and have no meaning at all, often resulting in spurious errors. To tackle this shortcoming, we propose the VeriFlow architecture as a flow based density model tailored to allow any verification approach to restrict its search to the some data distribution of interest. We argue that our architecture is particularly well suited for this purpose because of two major properties. First, we show that the transformation that is defined by our model is piecewise affine. Therefore, the model allows the usage of verifiers based on constraint solving with linear arithmetic. Second, upper density level sets (UDL) of the data distribution take the shape of an $L^p$-ball in the latent space. As a consequence, representations of UDLs specified by a given probability are effectively computable in the latent space. This property allows for effective verification with a fine-grained, probabilistically interpretable control of how (a-)typical the inputs subject to verification are.

## 1 Introduction

The outstanding performance of neural networks in tasks such as object detection (Zhao et al., 2019) image classification (Rawat & Wang, 2017), anomaly detection (Pang et al., 2021) and natural language processing (Goldberg, 2016) made them a popular solution for many real-world-applications, including safety-critical ones. With the increasing popularity of neural networks, defects and limitations of these systems have been witnessed by the general public. The AI incident database[1] keeps track of harms and near-harms caused by AI-Systems in the real world.

Ideally, safety and fairness properties of such inherently opaque neural networks should be formally guaranteed when used in safety-critical applications. As a solution, formal verification can be used to check whether a neural network satisfies a given safety property for the entire input space, or whether there exists some (synthetic) input for which the desired property is violated. This is in contrast to the statistical testing methods classically employed in Machine learning, where the output of the neural network is checked for a finite set of samples, usually from a held-out test set.

However, state-of-the-art formal verification methods only allow for verifying either global or local properties. Global properties ensure a specific behaviour of the neural network on the whole input space. As an example, fairness properties require the neural network to predict the same output for any two inputs that only differ in some sensitive attribute. Local properties, on the other hand, ensure a specific behaviour of the neural network only in some part of the input space that is usually restricted using the training set. One well-studied example for a local property is *adversarial robustness* which requires the neural network to classify any point from the data set as the same class as any minor perturbation of that point. However, both global and local properties have shortcomings limiting their applicability. Global properties refer to the entire input space, but we may not want or need to verify the property for noise-inputs or on regions of the input space for which there were no training samples available (epistemic uncertainty). Local properties, on the other hand, suffer from the same problem as statistical testing, i.e., they rely on a high-quality data set that the verification property is based on.

---

[1] https://incidentdatabase.ai/

To overcome these problems, we design a flow model tailored towards the application in neural network verification and leverage it to restrict the input space of the neural network under verification to the underlying data distribution. In contrast to generative adversarial networks (GANs) and variational autoencoders, flow models do not only allow for efficient sampling but also provide probabilistic interpretability via tractable likelihoods (Papamakarios et al., 2021; Goodfellow et al., 2014; Rezende & Mohamed, 2015; Dinh et al., 2015; Tabak & Vanden-Eijnden, 2010). This feature is important to facilitate fine-grained probabilistic control when restricting the input space to typical inputs when specifying the verification property. This approach makes the verification property less reliant on the dataset, while still keeping the input space focused on meaningful data.

To illustrate this idea, consider a neural network trained to classify images of handwritten digits. Assume we want the classifier to always be confident about it's classification of a certain digit. More precisely, our verification property is that the neural network's confidence is high if it classifies an image as 7 or 8, respectively. If the neural network does not satisfy the verification property, the verification tool will return an image that is classified as 7 or 8 with low confidence. Such counterexamples are illustrated in Figure 2. The two counterexamples on the left side in Figure 2 were found with a traditional (constraint-based) verifier without leveraging our flow model. These counterexample are noise images without any meaning and come from a region of the input space with high epistemic uncertainty. However, the counterexamples on the right side of Figure 2 were obtained by leveraging our flow model to restrict the input space of the verification property to typical inputs. In the context of Figure 2, the right side restricts the input space to a UDL of given probability and therefore, the counterexamples come from within the data distribution and provide better insights into the classifiers' weakness when used in a real-world scenario. We refer to Section 4 for more details on this experiment.

We briefly list our main theoretical results. We design a novel flow model with $L_1$-radially monotonic base distributions that provides the following theoretical properties crucial for the verification domain while outperforming its normalizing counterparts in nearly all benchmarks. Specifically, we design a flow model that:

1. Maps upper/lower (log-)density level sets of the target distribution to upper/lower (log-)density level sets of the base distribution allowing for fine-grained probabilistic control during sampling.

2. Allows the definition of the pre-image of a density level set in the latent space via linear constraints.

These properties enable us to restrict the verification to a probabilistically meaningful subset of the input space and differentiate VeriFlow from generic flow architectures. Indeed, density level sets of flow models generally do not have tractable pre-images in latent space.

We identify sufficient conditions for a flow layer to have the aforementioned properties, survey the literature, and present a collection of layers - with its most representative members being additive coupling layers and bijective affine layers - that can be arbitrarily combined to yield a flow with the desired properties.

## 2 BACKGROUND AND ORGANIZATION

Verifying neural networks involves checking if a network $f$ satisfies a semantic property $P$, often expressed as $\phi(x) \implies \psi(f(x))$, where $\phi$ and $\psi$ are pre- and postconditions. In this paper, we use two conceptually different verification approaches: *constraint-based verification* and *abstract interpretation*. We briefly explain the core idea of each verification approach and refer to Albarghouthi (2021) for an in-depth explanation.

**Constraint-based Verification** This approach translates $f$ and $P$ into a logical formula $\Psi_{f,P}$, whose validity implies $f$ satisfies $P$. Efficient SMT solvers like Marabou 2 (Wu et al., 2024) handle these formulas for networks with linear components such as ReLU activations. Verifying $\Psi_{f,P}$ involves checking the unsatisfiability of $\neg\Psi_{f,P}$. If $\neg\Psi_{f,P}$ is satisfiable, a counterexample exists. These methods are *complete* and provide counterexamples when the property is violated, though their runtime can be high for positive proofs.

**Abstract Interpretation**   Abstract interpretation symbolically executes a neural network using geometric abstract domains like zonotopes, which over-approximate input sets. By propagating these through the network, the procedure over-approximates possible outputs. Verification succeeds if the output lies entirely within the "safe space" defined by the semantic property $\psi$. If outputs are outside, the property fails; partial overlap leaves the result inconclusive due to over-approximation.

**Organization**   On a high level, we propose a flow model that piecewise linear and thus, can be encoded into the semantic property $P$ and used as part of the specification for the downstream verification task. In the next section, we present our flow model architecture. To the best of our knowledge, all propositions in Section 3 are novel contributions. In the consequent Section 4, we show our experimental results and showcase, how the flow model can be leveraged for enhancing the verification of a global property.

## 3   FLOW MODELS

Flow models provide an elegant way to represent a density estimator and a generative model by a single network. More precisely, we train a flow to transform a simple base (or latent) distribution $B$ into the data (or target) distribution $D$ using a continuous invertible map with continuous inverse, i.e. a diffeomorphism [2]. The map $F$ is implemented by an invertible neural network. We obtain a density estimator and a generative model by applying the flow in both directions:

1. Sampling is performed by first sampling $z \sim B$ and then computing the map $F(z)$.
2. The likelihood $p_D(x)$ is computable with the change of variables formula (Folland, 1999):

$$p_D(x) = \left| \det \frac{\partial F^{-1}}{\partial x^T} \right| p_B(F^{-1}(x)). \tag{1}$$

While most neural networks are intrinsically differentiable, they do not represent bijections in general. One needs to design specific architectures that restrict the hypothesis space to diffeomorphisms. Note that the existence of an inverse does not necessarily imply that the inverse can be easily computed. There are flow architectures that allow only one of the above operations to be efficiently performed (Kobyzev et al., 2020; Papamakarios et al., 2021). However, it is also not uncommon that both operation have the same complexity (Dinh et al., 2015). The major goal of this work is to design a flow architecture that does not only allow to perform both operations efficiently, but it also allows an efficient analysis of the flow in a verification context. For the latter, it is often required to not only sample individually, but verify on a space of sampled objects. We define the spaces containing high-density and low-density samples as upper- and lower density level sets.

**Definition 1.** *Given the density of the input distribution $p_D : \mathbb{R}^d \to \mathbb{R}_+$. The set of points whose density exceeds a given threshold $t$ is called the upper density level set (UDL) and is defined as $L_D^{\uparrow}(t) := \{x \in \mathbb{R}^d \mid p_D(x) > t\}$. Respectively, the lower density level set (LDL) contains the set of points deceeding the density threshold: $L_D^{\downarrow}(t) := \{x \in \mathbb{R}^d \mid p_D(x) \le t\} = \mathbb{R}^d \setminus L_D^{\uparrow}$. If for $q \in [0, 1]$ there is a unique UDL of $D$ with probability $q$, then we denote this set by $UDL_D(q)$.*

Upper density level sets naturally capture the center of the distribution while LDLs capture the tail. Note that the existence and uniqueness of the UDL with given probability is guaranteed if $P_D(\{x \mid p(x)_D = t\}) = 0$ for all $t > 0$, where $P_D$ denotes the probability induced by $p_D$, i.e. $P(x \in S) := \int_S p_D(x)dx$.

**Base Distributions**   An isotropic Gaussian is by far the most common choice for the base distribution of a flow model. In this case, we refer to the model as a normalizing flow. A normal distribution might seem to be the natural choice, but it is definitely not the only option. In fact, the Knothe-Rosenblatt Rearrangement Theorem (Knothe, 1957; Rosenblatt, 1952) guarantees that any two absolutely

---

[2] Note that we define the flows in the direction from base distribution to data distribution. This is inverse to the direction suggested by the common name "Normalizing Flow", but better suited for our analysis. Apart from that, the two definitions are equivalent.

continuous distributions can be transformed into one another via a diffeomorphism between their supports. In the next section, we will show that under certain conditions $p$-radial monotonic base distributions with $p \in \{1, \infty\}$, especially the Laplacian distributions, provide some merits that allow efficient analysis of our flow model in verification scenarios. In our case, it even turned out that changing the base distribution boosted the performance of the model and the stability of the training.

**Definition 2.** *Let $k \in \mathbb{N}_{>0} \cup \{\infty\}$ and let $X$ be a random variable over $\mathbb{R}^d$. We say that $X$ is $k$-radially distributed if there is a function $\hat{p} : \mathbb{R}_+ \to \mathbb{R}_+$ such that $p(x) = \hat{p}(|x|_k)$. If $\hat{p}$ is also strictly monotonically decreasing, then we say that $X$ is $k$-radial monotonic.*

Moreover, $k$-radial distributions are easily definable starting from the corresponding distribution of $k$-norms. In the following, let $V_k^d(r)$ be the hyper volume of the $L^k$-ball of radius $r$ in $\mathbb{R}^d$.

**Definition 3.** *Let $\rho : \mathbb{R}_+ \to \mathbb{R}_+$ be a probability density and $k \in \mathbb{N}_{>0} \cup \{\infty\}$. Then we call $R_{\rho,k,d}$ the $k$-radial distribution with $k$-norm distribution $\rho$ in $d$-dimensional space, which is given by the probability density function $p_{R_{\rho,k,d}}(x) = \rho(|x|_k) \left( \frac{\partial V_k^d(r)}{\partial r}(|x|_k) \right)^{-1}$.*

In other words, a $k$-radial distribution is completely determined by the distribution of the $k$-norm. Note that if $X$ is radial monotonic, it does not imply that $p_{|X|_k}$ is monotonic. For instance, if $X$ is a $d$-dimensional standard Gaussian, then $X$ is 2-radial monotonic but $|X|_2$ is $\chi(d)$ distributed and hence not monotonic for $d > 1$.

The following observation is crucial for our application: The density level sets of a $k$-radial monotonic distribution are $L^k$-balls. By choosing $r(q) := \text{quantile}_{|X|_p}(q)$, we obtain the following result:

**Proposition 1.** *Let $X$ be a $k$-radial monotonic random variable on $\mathbb{R}^d$. Then there is a function $r : [0, 1) \to \mathbb{R}_+$ such that for any $q \in [0, 1)$, the upper density level set of probability $q$ is given by $UDL_X(q) = \mathbb{B}_k^d(r(q))$, where $\mathbb{B}_k^d(r)$ is the $L^k$-ball of radius $r$ with center at the origin.*

**Piecewise Affine Transformations** A function $f : X \to Y$ is piecewise affine, if there is a partition of the domain $X = X_1 \cup \cdots \cup X_n$ such that $f$ restricted to $X_i$ is affine for all i. We call $X_1, \ldots, X_n$ affine regions of $f$. As we argued earlier, piecewise affinity is crucial for efficient SMT based verification. For the usage of our flow model this means that the transformation model should be piecewise affine. A natural approach to start off is therefore the use of piecewise affine networks, represented e.g. by ReLU networks. If we can ensure that the defined function is bijective, then we obtain a continuous piecewise affine bijection where the affine regions can be represented as intersections of open and closed half-spaces (Moser et al., 2022). Hence, the regions are contained in the Borel algebra $\mathcal{B}(\mathbb{R}^d)$. It is straight forward to show that the change of variables formula applies piecewise. We include a proof in the supplementary material for the sake of self-containedness. Intuitively, Proposition 2 states that the change of variables formula is valid for piecewise affine functions, if the affine regions are Borel sets and the determinant is computed piecewise.

**Proposition 2.** *Let $F : \mathbb{R}^d \to \mathbb{R}^d$ be a piecewise affine bijection with affine partition $R_1, \ldots, R_n \in \mathcal{B}(\mathbb{R}^d)$ of the input space and corresponding affine functions $f_1, \ldots, f_n$. Let $X$ be an absolutely continuous random variable. Then $p_{F(X)}(y) = p_X(F^{-1}(y)) \left| \det \frac{\partial F^{-1}}{\partial y} \right|$, where the Jacobian of $F^{-1}$ is evaluated piecewise. More precisely, $\left| \det \frac{\partial F^{-1}}{\partial y} \right| = \sum_i \left| \det \frac{\partial f_i^{-1}}{\partial y} \right| \cdot \mathbb{I}\left[ F^{-1}(x) \in R_i \right]$, where $\mathbb{I}$ is the indicator function.*

**Uniformly Scaling Flows** If we restrict the flow model to be uniformly scaling, i.e. demand that the Jacobian determinant of the transformation is constant, then we obtain an intriguing way of defining density level sets of the data distribution.

**Proposition 3.** *If the determinant of the Jacobian of a flow $F$ on $\mathbb{R}^d$ is constant, then $F$ maps upper density level sets of the target distribution to upper density level sets of the base distribution. Hence, if $B$ is a $k$-radial monotonic distribution over the domain of $F$, then there is a function $r : [0, 1) \to \mathbb{R}_+$ such that $UDL_{F(B)}(q) = F(\mathbb{B}_k^d(r(q)))$.*

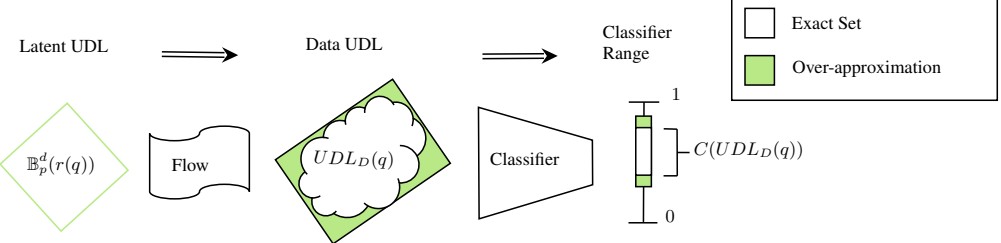

**Fig. 1:** Visualization of an abstract interpretation approach to in-distribution verification of a classifier using a flow model with constant Jacobian determinant and a $p$-radial monotonic base distribution. The procedure starts by defining the UDL exactly in the latent space. The true classification range w.r.t. the UDL equals the result of pushing the set consecutively through the flow and the classifier. An over-approximation can be obtained via abstract interpretation.

The proposition is especially attractive for abstract interpretation methods. For a radial monotonic base distribution w.r.t. the 1- or the $\infty$-norm, density level sets are definable by linear constraints. They can therefore act as an initial set that is propagated through the flow to obtain an approximation of the upper density level set of the data distribution. As we can derive from Proposition 1, the function $r$ is simply the quantile function of the $k$-norm distribution of $B$. Hence, Proposition 3 yields an effective way to define an upper density level sets with a given probability. The approach is summarized in Figure 1. A major challenge in the application, however, will be the tightness of the approximation of the non-linearity. There is a delicate trade off to be made between tightness of approximation and complexity of the description. Interesting work in this direction has been done by Bak (2021b).

Regarding SMT based methods, it might also be of interest whether the log-density function of the model is piecewise affine, since this would allow us to address the density freely within neuro-symbolic specifications. We mention here that it is indeed the case for piecewise affine flows and log-piecewise affine base densities such as the Laplacian.

Interestingly, it turns out that restricting the popular coupling layers to piecewise affine operations naturally leads to uniformly scaling flows. We shortly summarize our findings in the remaining of this section and refer to the extended architectural survey in the supplementary material for more information.

We survey the literature to identify learnable components that satisfy our needs. Most prominently, additive conditioning layers, like additive coupling, turn out to yield exactly the networks that we envision. This encompasses the seminal NICE architecture (Dinh et al., 2015), additive auto-regressive layers (Kingma et al., 2017; Huang et al., 2018; Papamakarios et al., 2017), and masked additive convolutions (Ma et al., 2019).

Additionally, bijective affine transformations represented by LU-decomposed affine layers (LUNets by Chan et al. (2023)) turn out to be equally well suitable. These layers also constitute a powerful replacement for the permutations or masks that are usually employed before a coupling-like layer. Although this addition turned out to be very beneficial, we note that the number of parameters scales quadratically with the dimension, which can be a performance bottleneck on high dimensional data.

Finally, we summarize the good properties of flows build from these layers in the following proposition, which also defines our proposed architecture. A more in-depth treatment of the layer architectures can be found in the supplementary material.

**Proposition 4 (VeriFlow).** *Let $F$ be a network that is purely built from the layer types (masked) additive coupling, additive autoregression, masked additive convolution, LU layers, component-wise scaling, and permutation of input dimensions. If the first three layer types only use piecewise affine conditioning networks, then $F$ is a uniformly scaling piecewise affine flow.*

## 4 EXPERIMENTS

The goal of our experiments is twofold. Firstly, in Section 4.1, we show that the VeriFlow architecture can be integrated with common verification frameworks for scalable verification and better counterexamples when unsafe. Secondly, in Section 4.2 we show that combining LU-layers with additive coupling greatly improves over the baseline performance of the NICE architecture and that certain 1-radial base distributions outperform their normalizing counterparts in the majority of benchmarks.

### 4.1 VERIFICATION EXPERIMENTS

We conduct our verification experiments on a downscaled version of the MNIST dataset, where the original $28 \times 28$ pixel images have been reduced to $10 \times 10$ pixels. This downsizing was necessitated by the limitations of the constraint-based verifier Marabou, which struggles to scale to large networks. It is essential to note that this constraint is specific to the verifier used and not an inherent restriction of our approach. In fact, our subsequent results demonstrate that by leveraging abstract interpretation, our methodology successfully scales to larger networks. Nonetheless, to maintain comparability across our results, we also used the downscaled MNIST dataset for our scalability experiments.

We trained a total of 10 flow models independently from each other. Each flow model is trained on a specific MNIST class representing a digit. We denote a flow model for digit $i$ on our downscaled MNIST dataset (MNIST$_i^{10 \times 10}$), which we denote by $g_0, \ldots, g_9$. Each flow model in our experiments has 3 additive coupling layers with 100 neurons and ReLu activation functions. Furthermore, we trained classifiers with varying depth $f_\ell$, where $\ell$ corresponds to the number of layers $1 \leq \ell \leq 15$, with each layer consisting of a matrix multiplication, addition and ReLU activation, in the network. We trained the networks with Adam-optimizer obtaining an accuracy score of around $90\%$ for all classifiers. We did not aim at making the classifiers particularly safe or unsafe w.r.t. the verification tasks at hand. The final classification then corresponds to the respective digit with the highest score.

As verification tools, we use the Python interface of the C++ implemented verification framework Marabou 2 (Wu et al., 2024) for deductive verification and ERAN for abstract interpretation (ERAN). Since both verifiers only allow for one network to be parsed, we merge both networks together by piping the output of the flow model $g_\tau$ into the classifier $f_\ell$, resulting in one neural network computing the composition $f_\ell \circ g_\tau : \mathbb{R}^{10 \times 10} \to \mathbb{R}^{10}$. We run all our verification-experiments on a Ubuntu 22.04 machine with an i7-1365U CPU at 1.80 GHz, 32 GB RAM and Intel Iris Xe Graphics.

**Use Case: Better Counterexamples** As a representative use case of our flow model, this section demonstrates the effectiveness of our approach in generating more realistic counterexamples during verification, as compared to those obtained without our model. To illustrate this, we consider a simple yet illustrative verification condition in which the classifier $f_1$ is required to have a high confidence on all images classified as a specific digit $\tau$ (the classifier $f_1$ was chosen randomly). Counterexamples to this property consist of images classified as $\tau$, yet with a confidence lower than $\delta$. Examples of such images are presented in Figure 2.

The top row of Figure 2 shows the preconditions, assignments, and postconditions used in our experiments. The postcondition $\psi$ is the same for all experiments: if the network $f_1$ classifies an input as class $\tau$, then its "confidence" is higher than a fixed threshold $\delta$. We follow Xie et al. (2022) and define a network's confidence as $conf(\boldsymbol{y}, \tau) := (|\boldsymbol{y}| \cdot \boldsymbol{y}[\tau] - \sum_{j \neq \tau} \boldsymbol{y}[j]) / |\boldsymbol{y}|$, where $|\boldsymbol{y}|$ denotes the number of elements in the output of the classifier $\boldsymbol{y}$. This particular notion of confidence is somewhat artificial, but it is useful in two regards: (i) it illustrates the shortcomings of traditional verification approaches when verifying a global property and (ii) it can easily be handled by existing constraint-based verifiers (e.g., Marabou 2) due to its piecewise linearity. Note, however, that our approach is not limited to this notion of confidence and can handle any verification condition that the underlying verifier can.

The results in Figures 2a and 2b differ due to the use of different preconditions. In Figure 2a the precondition restricts the input to be *any* grayscale pixel image with resolution $10 \times 10$. That image is then applied to the classifier $f_1$ to obtain the scores $\boldsymbol{y}$. In Figure 2b, on the other hand, the precondition restricts the input $\boldsymbol{x}$ to the UDL in the latent space. The threshold $t$ is determined such that $p_D(L_D^\uparrow(t_p)) = p$ where $p = 0.01$. Applying the flow $g_\tau$ yields a top $1\%$ typical image which is then applied to the classifier $f_1$ to obtain the score $\boldsymbol{y}$. In all experiments, the network does not satisfy the property and the images below correspond to the counterexamples provided by the solver.

As can be seen, the images on the left, which do not utilize a flow model, are noise and come from a region of the input space with high epistemic uncertainty. This provides almost no insight to the weakness of the classifier when used in real world. However, the images on the right, that do utilize a flow model, come from within the data distribution and provide a deeper insight to the classifier's weakness as the images indeed resemble the digits 7 and 8. As an alternative to using flow models, autoencoders can also yield similar visual results (Xie et al., 2022). However, autoencoders lack probabilistic interpretability and fine-grained probabilistic control of the input-space, which are integral features of our flow-model. More experiments are shown in the appendix: Section C.2 shows a verification task involving epistemic uncertainty, while Section C.3 illustrates how the confidence threshold in the postcondition affects the quality of counterexamples.

**Scalability**    In this experiment, we assess the scalability of our flow model in the verification domain. To this end, we consider both the deductive verification tool Marabou 2 (Wu et al., 2024) and the abstract interpretation tool ERAN (ERAN). In order to assess the scalability reliably, we focus on verifying properties that are satisfied by the neural network. Otherwise, the deductive verifier could terminate early after finding a counterexample while barely touching the search space. We verify for a classifier $f_\ell$ that for a subspace of the $1\%$ UDL, the network $f_\ell$ classifies the whole output space of the flow model $g_0$ restricted by the UDL as the digit 0. We show this property only for a subspace of the UDL because the full UDL does not constitute a zonotope as required when using the deepzono domain (Singh et al., 2018). One could eliminate this problem by either choosing an $\infty$-radial base distribution for the flow or by using abstract interpretation algorithms that can handle more general initial sets. However, we decided to live with this shortcoming since first experiments indicated that both the above mentioned approaches bear additional challenges in terms of quality of the fit and efficiency, respectively, that we cannot fully address in this work. For observing potential effects of the size of the neural network on the runtime, we conduct experiments with several classifiers varying in depth $f^1, \ldots, f^{15}$ as indicated on the X-Axis of Figure 3a and repeat each experiment three times, taking the median value. Note that the y-axes in Figure 3 are discontinuous.

The results in Figure 3a indicate that the runtime of both tools is linear in the depth of the neural network. And even though Marabou is slower by a factor of up to 70 compared to ERAN, both verifiers accomplish the verification tasks within seconds. In a second experiment, we fix the neural network to $f_1$ and only increase the size of the input space. The small space corresponds to a small percentage of the whole input space of around $1e^{-}7\%$. This input size was selected in order to illustrate the exact threshold in input size for which the runtime of verification with Marabou becomes prohibitive. We present the results of the second experiment in Figure 3b. Clearly, ERAN scales better than Marabou in both absolute values with a factor of around 50 for the smallest tested input size as seen before, as well as in the overall trend when increasing the size of the input space to search in. In particular, we can now also observe a strong non-linear increase in the runtime of Marabou, as it reaches the timeout limit of 200 seconds for an input size that is proved by the abstract interpreter within 20 milliseconds. We conjecture that it is because Marabou also uses abstract interpretation methods as a preprocessing step for inferring bounds for each node in the network and use these bounds to simplify or even trivialize the satisfiability problem. This, however, may no longer be

$$\phi\colon \left\{\boldsymbol{x} \in [0, 255]^{10 \times 10}\right\}$$
$$\boldsymbol{y} \leftarrow f_1(\boldsymbol{x})$$
$$\psi\colon \left\{argmax(\boldsymbol{y}) = \tau \rightarrow conf(\boldsymbol{y}, \tau) > \delta\right\}$$

$$\phi\colon \left\{\boldsymbol{x} \in L_D^{\uparrow}(t)\right\}$$
$$\boldsymbol{y} \leftarrow f_1(g_\tau(\boldsymbol{x}))$$
$$\psi\colon \left\{argmax(\boldsymbol{y}) = \tau \rightarrow conf(\boldsymbol{y}, \tau) > \delta\right\}$$

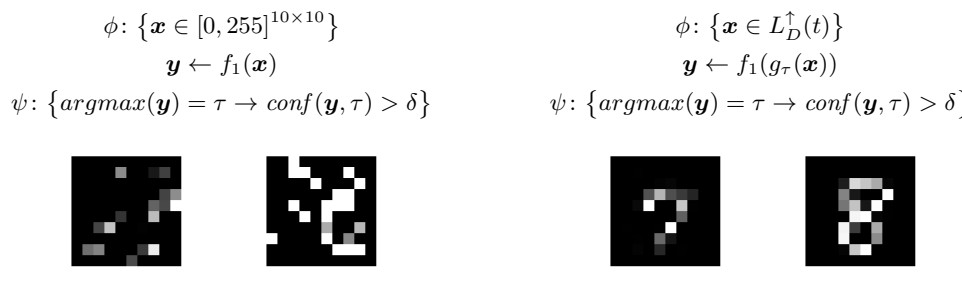

**(a)** Counterexamples without flow model          **(b)** Counterexamples with flow model

**Fig. 2:** The formulas at the top correspond to the verification conditions with $\tau = 7$ for each left side and $\tau = 8$ for each right side and $\delta = 17$. The images at the bottom are counter-examples as provided by the solver.

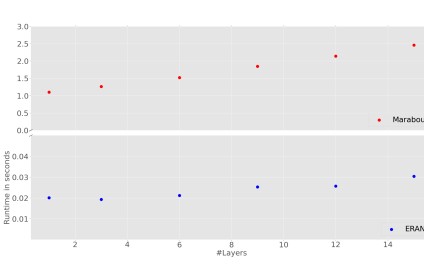

**(a)** Comparison increasing depth of the classifier

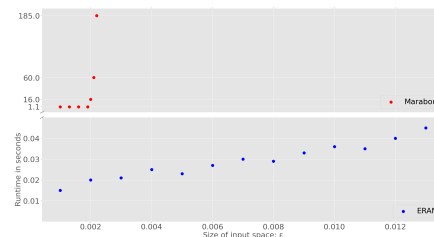

**(b)** Comparison increasing input sizes

**Fig. 3:** Comparison of the runtime between ERAN (blue) and Marabou (red) for a complete proof.

feasible when the input space gets overly increased. We conclude that both ERAN and Marabou scale well for deeper neural networks and ERAN also scales well for increased input sizes.

**Unscaled MNIST** In our previous experiments, we used the rescaled MNIST in order to directly compare the runtime of the verifiers ERAN and the computationally more expensive verifier Marabou. (Brix et al., 2023) Now we demonstrate that the training procedure of our flow model scales to higher dimensional datasets such as MNIST $28x28$. We trained the flow model the same way as with MNIST 10x10. The the quality of random samples from the flow model are shown in Figure 4a. The total number of parameters of the flow model increased by a factor of 22 compared to the flow models on MNIST $10x10$ and the number of parameters of the classifier increased by a factor of 20. On the verification side, we only show the runtime of the ERAN verifier for this larger flow model as Marabou times out after 60 minutes in our experiments. The runtime results for the bound-propagation algorithm of ERAN are plotted in Figures 4b. The runtime of ERAN increased by a factor of ten compared to the flow model for MNIST $10x10$. The composed neural network in Figure 4b has approximately 2.9M parameters (110K of which are due to the classifier).

## 4.2 ABLATION STUDY

Besides the efficient applicability of our model in neuro-symbolic verification, our model needs to be versatile enough to capture the concepts of interest. We show the effectiveness of our architecture as density estimator and generative model. We perform several ablation studies to compare our choice of the base distribution against the most commonly used Gaussian distribution as well as our architecture against the original NICE architecture. Our most surprising finding is that 1-radial base distributions are not only competitive but outperform their Normal counterparts in the vast majority of benchmarks. Overall, we observed that the training with an 1-radial base distribution, especially a Laplacian, is more stable (note the bad performance and high standard deviations for some digits with the normal distribution in Figure 5 (left)).

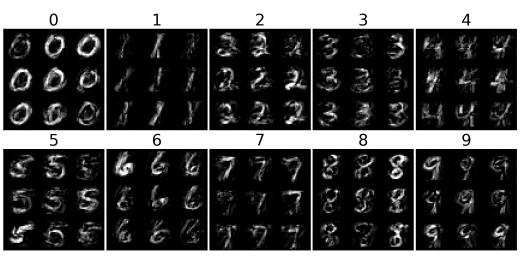

**(a)** Full MNIST Random samples

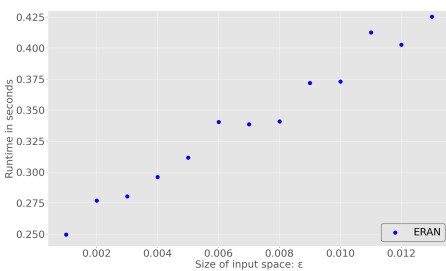

**(b)** Runtime effect of increasing Input space

**Fig. 4:** Training and verification results with full MNIST 28x28

As benchmarks, we mostly focus on tasks of moderate dimensionality and small sized networks since this scenario is approachable with the contemporary verification software on standard hardware. We pick up the example from the verification experiments and fit the models to each individual MNIST digit $i$ where the images are rescaled to $10 \times 10$ pixels ($\text{MNIST}_i^{10 \times 10}$) and we use three classic synthetic 2D datasets (circles, moons, blobs). All these experiments have been performed on a Macbook pro with M2 chip and 16GB of RAM. Additionally, we also scale our architecture to higher dimensions and more challenging datasets by performing a base distribution comparison on the full MNIST. Performance metric in all experiments is the negative log-likelihood (NLL). The scaled experiments have been performed on a DGX2 with 8 V100 GPUs. For all image datasets we used uniform dequantization and report the NLL of dequantized images.

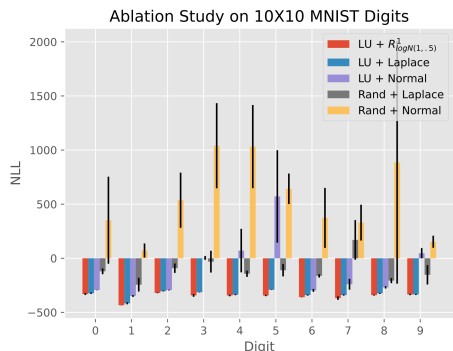

| Dataset | Base Distribution | NLL |
|---|---|---|
| MNIST | Normal | -1679.132 |
| | Laplace | -1879.869 |
| | $\mathbf{R_{logN(1, .5),1,784}}$ | **-2013.079** |
| Circles | Normal | -0.910 |
| | **Laplace** | **-1.113** |
| | $R_{\log N(1, .5), 1, 2}$ | -0.604 |
| Moons | Normal | -1.899 |
| | **Laplace** | **-2.279** |
| | $R_{\log N(1, .5),1,2}$ | -1.084 |
| Blobs | **Normal** | **2.287** |
| | Laplace | 2.500 |
| | $R_{\log N(1, .5),1,2}$ | 2.596 |

**Fig. 5:** (**Left**) Ablation study on $\text{MNIST}_i^{10 \times 10}$. We fit our flow architecture with Normal and Laplacian base distributions and additive coupling layers, which are alternated either with random masking layers or with LU-layers. We perform a random hyperparameter optimization with 20 samples for each configuration and report the average top-3 performance. The remaining experiment parametrization is fixed across all models and datasets. We use the average negative log-likelihood on the test set (lower is better) as performance measure. (**Right**) Negative log-likelihood (lower is better) of VeriFlow with varying base distribution on multiple benchmark datasets. The architecture is always based on alternating LU layers and additive coupling layers. The remaining experiment parametrization is fixed per dataset.

## 5 RELATED WORK

We first provide an overview of verification tools that can be leveraged for verification and could take our flow model as part of the specification. Generally speaking Polytopes (Chen et al., 2008), Zonotopes or even simple boxes are sufficient for representing the UDL of our flow model in the latent space precisely. This enables the use of more recent developments based on these domains that enhance precision or scalability for verification such as Deepzono (Singh et al., 2018), DeepPoly (Singh et al., 2019a), GPUPoly (Serre et al., 2021), RefineZono (Singh et al., 2019b), multi-neuron abstraction (Müller et al., 2023) and DiffPoly (Banerjee et al., 2024).

Besides of Marabou and ERAN, another verification framework that achieves promising results in the VNNComp competition is $\alpha, \beta-$crown (abcrown). It consists of numerous verification algorithms and combines abstract interpretation methods with branch-and-bound methods. In particular, GCP-CROWN (Zhang et al., 2022) recently became part of the $\alpha, \beta-$crown framework and enables the use of general cutting plane methods in combination with GPU accelerated bound propagation methods. Similarly, the verifier MN-BAB (Ferrari et al., 2022) is utilizes both branch-and-bound and convex relaxation but still provides *completeness* of the verification result.

Other verifiers that participate in the VNNComp are Cora Althoff (2015), PyRAT (Lemesle et al., 2024), nnenum (Bak, 2021a) and NNV (Lopez et al., 2023). We refer to the VMNComp (Brix et al., 2023) competition for a comprehensive overview. Note that while the aforementioned verifiers are conceivable for use as downstream verifiers with VeriFlow, they are generally *incomplete* and do not provide counterexamples when the neural network is unsafe.

However, the research in the verification context often focuses on the verification of local robustness properties (Balunovic et al., 2019; Zeng et al., 2023; Banerjee & Singh, 2024). These works also

uncover that neural networks are highly non-robust even for small perturbations and even when trained with robust training algorithms (Gowal et al., 2019; Zhang et al., 2019b;a). Verification of global properties as tackled in our work is arguably harder as it is naturally subject a greater input space. In this context, our flow model enables relaxing the global property by restricting the input space to the high-density region of the data distribution. This removes the necessity of the neural network to ensure the global property on the whole input space, which includes meaningless noise data. In other words, VeriFlow aims to push the challenging verification of global properties more towards local properties by restricting the input space to the data distribution.

Optimization-based approaches have recently emerged as a more scalable and higher-performing alternative to constraint-based and simple zonotope-based methods for verifying neural networks (Toledo et al., 2021; Wu et al., 2023; Mangal et al., 2020; Müller et al., 2023; Koller et al., 2024). While our flow model is, in principle, compatible with these optimization-based approaches and our restriction of the search space to an upper density sets can be formulated as a constrained optimization, we leave a thorough evaluation of their integration for future work.

Paralleling our efforts to design a flow model amenable to verification using existing infrastructure, recent studies have explored automated preprocessing techniques for neural networks, including pruning (Guidotti et al., 2020) and regularization methods (Leofante et al., 2023; Böing & Müller, 2022). The ultimate goal of this direction is to render them more suitable for verification with state-of-the-art verifiers.

Flow models are on the forefront of modern density estimation techniques and have received significant attention over the last decade (Papamakarios et al., 2019). A constant Jacobian determinant is usually observed at the time of the introduction of the respective layer in the context of the likelihood computation (Dinh et al., 2015; Ma et al., 2019; Kingma et al., 2017; Huang et al., 2018; Papamakarios et al., 2017), although typically without further investigation of the induced properties. The role of the Jacobian determinant in general has been investigated in the context of the exploding determinant phenomenon (Kim et al., 2020; Liao & He, 2021; Lyu et al., 2022). There is also a notable application of flow architectures with constant Jacobian determinant for anomaly detection. OneFlow uses a constant Jacobian determinant to compute and minimize the volume the image of a unit hyper sphere around the origin in the latent space, drawing a connection between flows and deep one-class SVMs (Maziarka et al., 2022).

## 6 CONCLUSION

We have presented the VeriFlow architecture, a flow-based density model that enables effective verification of neural networks within a specified data distribution and fits in any existing verification infrastructure. By using a novel approach of restricting the search space to probabilistically meaningful subsets of the input space, VeriFlow mitigates spurious errors and provides fine-grained, interpretable control over the input space. Independent of the verification domain, VeriFlow outperforms the NICE architecture, taken as baseline for uniformly scaling flow architectures, by a large margin.

Our flow-based verification approach shares a limitation inherent to neuro-symbolic frameworks: the quality of the verification results is contingent upon the quality of the specification networks, in this case, the flow model. However, as noted by Xie et al. (2022), these networks are generally smaller and more manageable than the networks under verification, allowing for additional training efforts and quality assurance measures, such as adversarial training. Moreover, the creation and improvement of flow models could be facilitated through public competitions, community-driven initiatives, or even future regulatory oversight, ultimately ensuring their accuracy and reliability.

We envision three promising avenues for future research. Firstly, we believe that verification tools should be enhanced to provide more comprehensive support for the capabilities of popular deep learning frameworks like PyTorch, particularly in scenarios involving multiple networks. Secondly, a deeper theoretical and practical understanding of the expressivity of piecewise affine uniformly scaling flows is needed. To the best of our knowledge there is no known universal approximation theorem that applies to our architectures. Lastly, well-calibrated density level sets are essential for producing interpretable verification results. Therefore, improving the consistency of density level sets in deep learning models poses an important challenge that warrants further investigation.

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

## A    PROOFS OMITTED FROM SECTION 3

**Proposition 2.** *Let $F : \mathbb{R}^d \to \mathbb{R}^d$ be a piecewise affine bijection with affine partition $R_1, \ldots, R_n \in \mathcal{B}(\mathbb{R}^d)$ of the input space and corresponding affine functions $f_1, \ldots, f_n$. Let $X$ be an absolutely continuous random variable. Then $p_{F(X)}(y) = p_X(F^{-1}(y)) \left| \det \frac{\partial F^{-1}}{\partial y} \right|$, where the Jacobian of $F^{-1}$ is evaluated piecewise. More precisely, $\left| \det \frac{\partial F^{-1}}{\partial y} \right| = \sum_i \left| \det \frac{\partial f_i^{-1}}{\partial y} \right| \cdot \mathbb{I} \left[ F^{-1}(x) \in R_i \right]$, where $\mathbb{I}$ is the indicator function.*

*Proof.* We define the random variable

$$C : \mathbb{R}^d \to \{1, \ldots, n\}; x \mapsto \sum_{k=1}^n k \cdot \mathbb{I}\left[ x \in R_k \right]$$

and consider the conditional probability densities $p_X(x \mid C = i) = P(X \in R_i)^{-1} p_X(x) \cdot \mathbb{I}[x \in R_i]$. Since $F$ is an affine bijection on $R_i$, the support of $p_X(\cdot | C = i)$, we can employ the change of variables formula and obtain that $p_{F(X)}(y \mid C = i) = P(X \in R_i)^{-1} p_X(f_i^{-1}(y)) \left| \det \frac{\partial f_i^{-1}}{\partial y} \right| \cdot \mathbb{I}[F^{-1}(y) \in R_i]$. Finally, we obtain by the sum rule that

$$p_{F(X)}(y) = \sum_{i=1}^n P(C = i) p_{F(X)}(y \mid C = i)$$

$$= \sum_{i=1}^n P_X(X \in R_i) p_{F(X)}(y \mid C = i)$$

$$= \sum_{i=1}^n p_X(f_i^{-1}(y)) \left| \det \frac{\partial f_i^{-1}}{\partial y} \right| \cdot \mathbb{I}[F^{-1}(y) \in R_i]$$

$$\overset{*}{=} \sum_{i=1}^n \left( \underbrace{\sum_{j=1}^n p_X(f_i^{-1}(y)) \mathbb{I}[F^{-1}(y) \in R_j]}_{= p_X(F^{-1}(y))} \right) \left| \det \frac{\partial f_i^{-1}}{\partial y} \right| \cdot \mathbb{I}[F^{-1}(y) \in R_i]$$

$$= p_X(F^{-1}(y)) \sum_{i=1}^n \left| \det \frac{\partial f_i^{-1}}{\partial y} \right| \cdot \mathbb{I}[F^{-1}(y) \in R_i]$$

$$= p_X(F^{-1}(y)) \left| \det \frac{\partial F^{-1}}{\partial y} \right|,$$

where $(*)$ holds since $\mathbb{I}[F^{-1}(y) \in R_i] \cdot \mathbb{I}[F^{-1}(y) \in R_j] = \delta_{ij} \mathbb{I}[F^{-1}(y) \in R_i]$, where $\delta_{ij} = \begin{cases} 1; i = j \\ 0; \text{ else} \end{cases}$ is the Kronecker-Delta. $\square$

Next, we consider the choice of the base distribution.

**Proposition 5.** *Let $p_D$ be defined by a piecewise affine flow $F$ and a log-piecewise affine base distribution $p_B$. Then $\log p_D$ is piecewise affine.*

*Proof.* As we have seen,

$$\log p_D(\mathbf{x}) = \log p_B(F(x)) + \log \left| \det \frac{\partial F}{\partial x} \right|$$

Since $F$ and $\log p_B(\cdot)$ are piecewise affine, $\log p_B(F(x))$ is also piecewise affine. Similarly, $\left| \det \frac{\partial F}{\partial x} \right|$ is piecewise constant, which implies that $\log \left| \det \frac{\partial F}{\partial x} \right|$ is piecewise constant too. The claim follows immediately. $\square$

**Proposition 3.** *If the determinant of the Jacobian of a flow $F$ on $\mathbb{R}^d$ is constant, then $F$ maps upper density level sets of the target distribution to upper density level sets of the base distribution. Hence, if $B$ is a k-radial monotonic distribution over the domain of $F$, then there is a function $r : [0,1) \to \mathbb{R}_+$ such that $UDL_{F(B)}(q) = F(\mathbb{B}_k^d(r(q)))$.*

*Proof.* This is a direct consequence of the change of variables formula.

$$F(\{x \mid \log p_D(x) > t\}) = \{F(x) \mid \log p_D(x) > t\}$$

$$= \left\{ F(x) \mid \log p_B(F(x)) > t - \underbrace{\log \left| \det \frac{\partial F}{\partial x} \right|}_{\text{const}} \right\}$$

$$= \{y \mid \log p_B(y) > t'\},$$

which is obviously an upper log-density level set w.r.t. the latent distribution $B$. The last equation holds since $F$ is a bijection and $\log \left| \det \frac{\partial F}{\partial x} \right|$ is constant. We combine the observation with the idea from 1 and conclude that for radial monotonic $B$ and $r(q) = \text{quantile}_{|B|_k}(q)$ the identity $UDL_{F(B)}(q) = F\left(\mathbb{B}_k^d(r(q))\right)$ is indeed correct. □

# B    AN EXTENDED ARCHITECTURAL SURVEY

## B.1    ADDITIVE TRANSFORMATIONS

As it turns out, additive transformations yield precisely the properties that we need in order to guarantee the good properties of the previous section. The simplest such architecture is realized by so called additive coupling, which was first introduced for the NICE architecture by Dinh et al. (2015).

## B.2    ADDITIVE COUPLING (NICE)

NICE belongs to the first flow architectures. Nevertheless, it is a popular benchmark which has shown good performance on multiple data sets.

**Additive Coupling Layers**    A NICE flow is build from additive coupling layers. Each such layer $L$ consists of a partition $I_1, I_2$ of $[D]$, where $D$ is the data dimension, and a conditioning function $m : \mathbb{R}^d \to \mathbb{R}^{D-d}$, where $d = |I_1|$. The layer $L$ maps $x$ to $y$ where

$$y_{I_1} = x_{I_1}$$
$$y_{I_2} = x_{I_2} + m(x_{I_1}).$$

It is easy to see that the Jacobean $\frac{\partial y}{\partial x} = \begin{pmatrix} I_d & 0 \\ \frac{\partial y_{I_2}}{\partial x_{I_1}} & I_{D-d} \end{pmatrix}$ is triangular and that all entries on the diagonal are 1. As the first $d$ components of the input remain unchanged, it is usually necessary to employ multiple layers with varying partitions of the input vector. It is straight forward to see that a coupling layer defines a piecewise affine function if the conditioner $m$ is piecewise affine.

**Allowing Rescaling**    As all additive coupling layers have Jacobean determinant 1, the same will hold for their composition. That means the space is never stretched or compressed through the transformation, which potentially limits the expressiveness. In order to account for this issue, NICE allows for a final component-wise rescaling, i.e. multiplication with a matrix $S$, where $S_{ij} \begin{cases} \neq 0 & \text{if } i = j \\ = 0 & \text{else} \end{cases}$.

**Computing log-Densities**    Because of the simple additive coupling, computing log-densities is particularly simple. Let $F$ be a NICE flow with base distribution $B$, layers $L_1, \ldots, L_n$, and scaling

matrix S. Then

$$\log(p_D(\mathbf{x})) = \log\left(p_B(F(x))\left|\det\frac{\partial F}{\partial x}\right|\right)$$

$$= \log\left(p_B(F(x))\cdot\prod\left|\det\frac{\partial L_i}{\partial x}\right|\cdot|\det S|\right)$$

$$= \log p_B(F(x)) + \underbrace{\sum\log\left|\det\frac{\partial L_i}{\partial x}\right|}_{=0} + \log|\det S|$$

$$= \log p_B(F(x)) + \sum\log S_{ii}$$

Computing $F^{-1}(z)$ has exactly the same complexity as computing a forward pass $F(x)$. Because in order to invert the flow we only need to multiply with the inverse scaling matrix and then pass the input to through the inverse coupling layer in reverse order. Note that for an additive coupling layer $L = ((I_1, I_2), m)$ the inverse function can be implemented by $L^{-1} = ((I_1, I_2), -m)$.

### MASKED ADDITIVE COUPLING

It is also possible to rewrite the additive coupling equation in order to implement the NICE architecture as a fully connected neural network with masking and skip connections. An additive coupling layer $\ell : \binom{x_{I_1}}{x_{I_2}} \mapsto \binom{x_{I_1}}{x_{I_2}+c(x_{I_1})}$, whose conditioner is implemented by a neural network $c$ can equivalently be written as

$$\ell(x) = x + (1 - \text{mask})\cdot c'(\text{mask}\cdot x), \tag{2}$$

where mask is a $\{0,1\}$-vector with $\text{mask}_i = 1 \Leftrightarrow i \in I_1$ and the multiplication is computed component wise. Further, $c'$ is a fully connected network obtained by adding dummy inputs for the components in $I_2$ and dummy outputs for the components of $I_1$ to $m$, which are effectively eliminated by the mask in Equation 2.

### ADDITIVE AUTO-REGRESSION

A general way to increase the expressiveness of the based flow models is the use of auto-regression instead of coupling (Kingma et al., 2017; Huang et al., 2018; Papamakarios et al., 2017). In this case the conditioner is implemented as an RNN $c$, which couples the input component by component. More precisely, an additive auto-regressive flow layer $\ell$ computes a transformation $\ell(x) = y$ with

$$h_1, z_1 = 0; \quad (h_{i+1}, z_{i+1}) = c(x_i, h_i)$$
$$y_i = x_i + z_i$$

Observe that the structure of the auto-regression still leads to a lower triangular shape of the Jacobean and the additive auto-regressive coupling ensures that all diagonal entries are 1. With these properties one easily checks Proposition 2, 5 and 3 remain valid if additive coupling is replaced by additive auto-regression.

### MASKED ADDITIVE CONVOLUTIONS

The idea of masking was used by Ma et al. (2019) in order to transfer coupling to convolutional architectures where the input is a higher-order tensor. We can also employ this idea in our situation and still maintain the desired properties. In this case, Equation 2 is applied to a convolutional network, e.g. with a checker board and/or a channel-wise mask. As the reader readily verifies, the analogues of Proposition 2 - 3 hold also for this layer.

### GOOD PROPERTIES OF ADDITIVE TRANSFORMATIONS

Let us summarize the properties of the above mentioned layers.

**Proposition 6.** *Let $F$ be a network that is purely build from the layer types (masked) additive coupling, additive autoregression, masked additive convolution, component-wise scaling, and permutation of input dimensions. If all conditioners are piecewise affine, then $F$ is a piecewise affine flow*

*with constant Jacobean determinant. In particular, any density $p_D$ defined by $F$ has the following properties:*

1. *If $B$ is the standard Laplacian distribution, then $\log p_D$ is piecewise affine*

2. *For any $p$-radial monotonic base distribution $B$ there is a function $r : [0, 1) \to \mathbb{R}$ such that $UDL_{F(B)}(q) = F(\mathbb{B}_k^d(r(q)))$.*

3. *Computing log-densities has the same computational complexity as sampling.*

### LUNets

Recently, bijective fully-connected layers have been proposed by Chan et al. (2023) as a so-called LUNet. The idea is to ensure that that both, the affine transformation of a fully connected layer and the non-linearity are bijections. Bijectivity is ensured by representing the linear transform of the layer by an LU-factorization $A = LU$ with lower/upper triangular Matrices $L$ and $U$. Bijectivity is ensured by adding the constraints that the diagonal of $L$ contains only ones and diagonal of $U$ is always non-zero. In this case, Propositions 2 and 5 will still hold if we replace the layer architecture and use leaky ReLU instead of ReLU, but Proposition 3 will in general not hold anymore as the determinant of the layer Jacobean is not constant anymore.

LUNet is a very different approach to guaranteeing the bijectivity of the transformation compared to additive coupling. It has the advantage that the entire input can be transformed by a single layer. The restriction that the affine transform needs to be bijective, however, fixes the capacity of the transformation to $d^2$ parameters where $d$ is the input dimension. This can be problematic, especially when working with high-dimensional data.

### Bijective Affine Layers

The bijective affine transform $T(x) = (LU)x + b$ at the heart of an LU-layer deserves special attention. Note that the determinant of the Jacobean is is constant for $T$. Since computing the inverse of an affine transform also has the complexity of computing the affine transform it follows that we can add bijective affine layers to the list of layers in Proposition 6 without loosing the validity of the statement. Bijective affine layers can be an interesting alternative to the intermediate permutation layers of the NICE architecture. Using an affine bijection instead of a simple fixed permutation of the dimensions allows the architecture to correlate the components of $I_1$ and $I_2$ in the subsequent coupling layer in a learnable fashion. As an example, consider the extreme case where all components of the target distribution are independent. In this case, the components $I_1$ and $I_2$ will be independent, no matter which permutation of the components we have applied beforehand. An affine bijection, however, can be capable of combining the variables in a way such that the components $I_1$ and $I_2$ become correlated.

**Proposition 7.** *The statement of Proposition 6 remains valid even if we add bijective affine transformations to the list of allowed layer types.*

## C   Experimental Setup

### C.1   Regularization and Advanced Training Methodology

Following the description given by Chan et al. (2023), we regularize the parameters of the LU layers. Without any form regularization, we observed exploding determinants on many tasks when working with LU layers. Additionally, we adopt the technique of soft training (Kim et al., 2020). During training, We sample a noise scale $\sigma$ from a prior distribution $P$ for each training sample and perturb the sample with noise sampled from from the base distribution (Gaussian or Laplacian) with covariance $\sigma I$. For the noise scale prior we use a Laplacian with small standard deviation. We fit a conditional flow on the perturbed data where the conditioning variable is the noise scale $\sigma$. During inference with unperturbed data, the noise scale is set to 0. We observed significant improvements through soft training, in terms of test likelihoods but also in terms of subjective visual quality as evaluated by the authors.

## C.2 Epistemic Uncertainty Verification

For the far tail of the data distribution there are usually no samples available. Hence, any model trained purely from data has never gotten information about these areas (epistemic uncertainty). In that context, we can verify that a classifier was trained with a vanishing inductive bias by moving away from the training data. In this case the uncertainty estimates given by a classifier should converge towards a prior distribution, e.g., uniform, as we move further outwards in the tail (Kendall & Gal, 2017; Hüllermeier & Waegeman, 2021). However, it is known that many deep neural network training methods produce badly calibrated networks with overconfident predictions, especially in areas of high epistemic uncertainty (Guo et al., 2017; Minderer et al., 2021). In that context, we want to verify that our classifier is not overconfident in the far tail of the data distribution. More precisely, we leverage our flow model to restrict the input to the $9\%$ tail of the data distribution, trimming the last $1\%$ to avoid an unbounded input space and verify that the classifier has a low confidence for all atypical inputs.

Similar to the in-distribution verification task, we conduct four experiments to compare the counterexamples without and with leveraging a flow model for restricting the input space and and present them in Figure 6 along with the verification properties.

The left upper side of Figure 6 is the same as for the in-distribution verification task, except for the postcondition $\psi$, that now checks for the confidence to be *low*, if the image is classified as the digit $\tau$. In the right upper side of Figure 6, the precondition $phi$ restricts the input space to the top one percent of most typical examples by first determining the threshold $t$ such that $p_D(L_D^\uparrow(t_p)) = p$ where $p = 0.01$ and setting the precondition $\phi\colon \{\boldsymbol{x} \in L_D^\uparrow(t)\}$. The postcondition on the right side $\psi$ checks for the confidence to be low, if the image is classified as $\tau$.

The counterexamples on the left side of Figure 6, that do not utilize a flow model, are noise-images that do not resemble even atypical digits, despite being classified with high confidence. The counterexamples on the right side of Figure 6, however, are atypical images of the digits that are classified with high confidence. The latter is more useful for a user as it shows exactly the type of images where the classification itself is reasonable but the high confidence shows a wrong calibration of the neural network.

$$\phi\colon \left\{\boldsymbol{x} \in [0, 255]^{10 \times 10}\right\} \qquad\qquad \phi\colon \left\{\boldsymbol{x} \in L_D^\downarrow(t)\right\}$$
$$y \leftarrow f(\boldsymbol{x}) \qquad\qquad\qquad y \leftarrow f(g_\tau(x))$$
$$\psi\colon \left\{argmax(y) = \tau \rightarrow conf(y, \tau) \leq \delta\right\} \qquad \psi\colon \left\{argmax(y) = 0 \rightarrow conf(y, \tau) \leq \delta\right\}$$

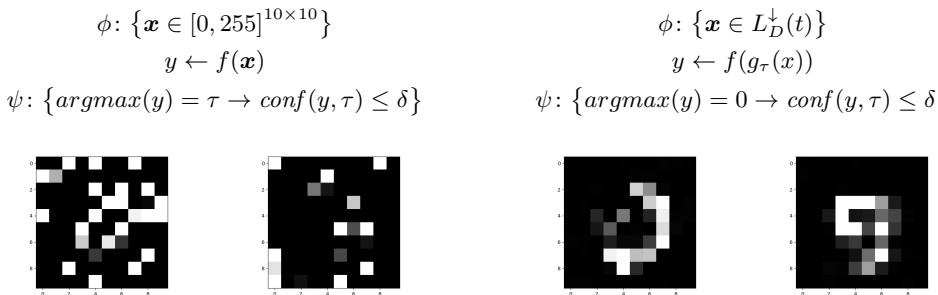

**Fig. 6:** The formulas at the top correspond to the verification conditions with $\tau = 0$ for each left side and $\tau = 9$ for each right side and $\delta = 8$. The images at the bottom are counter-examples as provided by the solver. Note that for obtaining the images on the right, the assignment $x$ is reapplied to the flow $g_\tau(x)$.

## C.3 Deductive Verification confidence threshold

One aspect that influences the quality of the counter examples of the verifier is the selected confidence threshold in the verification property. More precisely, running an in-distribution verification task on the same UDL but with different confidence thresholds for the classifier may also return more atypical images as shown in Figure 7. There, the UDL in the precondition is the same in every experiment, only the confidence threshold $\delta$ in the postcondition $argmax(y) = \tau \rightarrow conf(y, \tau) > \delta$ is assigned values increasing from 1 to 15 (with gaps in between).

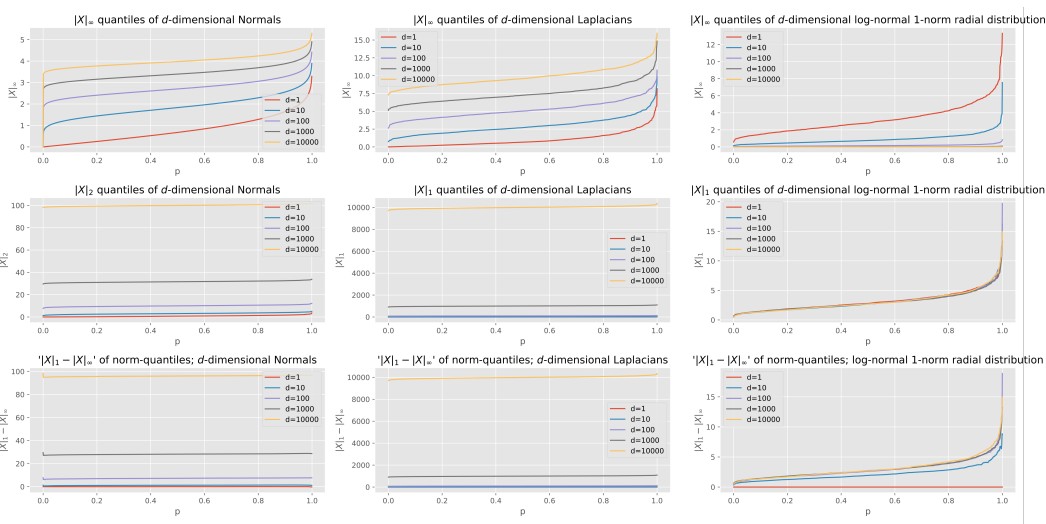

**Fig. 7:** Counterexamples for in-distribution verification tasks with increasing confidence thresholds in the postcondition.

### C.4   CALIBRATION OF DENSITY LEVEL SETS

A major challenge that we faced when conducting verification experiments was the calibration of the density level sets. When testing for satisfiability within a given density level set, we turn the distribution of interest into an uncertainty set in the latent space without preference for more likely examples. Current solvers tend to produce counter examples from extreme points within the uncertainty set. Since the geometry of the space often enforces that little probability is centered around these areas, we found that sampling from such point in the latent space often produces OOD data, even when considering a set that is supposed to represent the top e.g. one percent of most typical examples. Following this intuition, it seems that the properties of distributions like the Laplacian and Gaussian distribution in high dimensions lead to particularly unfavorable results for our purposes. Indeed, in high dimensional spaces, the corresponding $p$-norm distributions are strongly concentrated around the relatively large values of $d$ and $\sqrt{d}$, respectively. Therefore, we

**Fig. 8:** Quantiles of the $p$-norm distributions for $d$-dimensional Gaussian, Laplacian, and a custom 1-radial distribution. While the $\infty$-norm is always relatively low, the $p$-Norm of the $p$ radial distributions Laplacian and Gaussian are relatively large. That means that even on very high density contours, there are points on contour (e.g. $\alpha e_i$ for a standard basis vector $e_i$ and suitably chosen $\alpha$) that are very far away from the data that is seen during training (assuming that the empirical latent distribution approximately follows that base distribution relatively soon during training). Hence, it is not surprising that sampling from such areas in the latent space is likely to produce poor quality samples. The custom radial distribution mitigates this effect by keeping the $p$-norm distribution constant without dependency on the dimension.

conjecture that more concentrated base distributions help to mitigate this issue to some extend, see Figure 8 for an intuition. In order to avoid that the infinity norm of vectors becomes to small by concentrating the probability mass closer to the origin, we choose a unimodal radius distribution where the density converges to $0$ when approaching both, $0$ and $\infty$. Among our initial trials with multiple such distributions, like e.g. scaled Normal and Laplacian or distributions based on EVT norm distributions, a log-normal distribution with $\mu = 1.0$ and $\sigma = 0.5$ has shown most promising results. A qualitative and quantitative comparison with the Laplace base distribution is shown in the figures 9 – 11. We remark here that, strictly speaking, the resulting distribution is not radial monotonic. However, in high dimension it holds that the corresponding function $g$ with $g(|x|_1) = p(x)$ is unimodal with

it's mode extremely close to $0$. In an SMT setting, it would not be very hard to correct the formulas and exclude the small $L^p$ ball that does not belong to the density level set, but in our experiments with a dimensionality of 100, the additionally included area is so vanishingly small, both in probability and in volume, that we decided ignore this issue here. Based on our initial experience with more exotic radial distributions, we believe that a thorough investigation of this class of distributions is potentially interesting also in other application areas such as anomaly detection. More generally, the calibration of density level sets remains a challenge that we think has gotten too little attention in past. Therefore, we stress the need for more systematic research in that area.

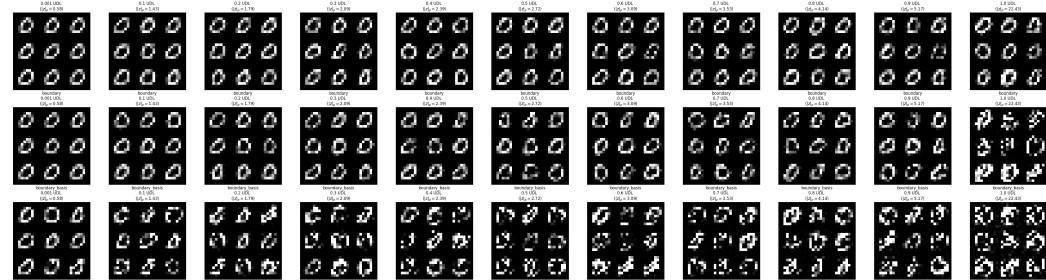

**Fig. 9:** Laplace base distribution

**Fig. 10:** Base distribution with Log-Normal 1-norm distribution.
Samples from two models trained on $\text{MNIST}_0^{10 \times 10}$. Samples are drawn from different region of the latent space. Each column considers a UDL of a given probability. The first row samples conditioned on being in the UDL. The second row samples uniformly from the density contour in the latent space, and the third row samples samples uniformly from the density contour in the latent space intersected with the union of the 1-dimensional subspaces induced by the standard basis vectors. Note that counter examples or often preferably chosen from the latter.

## C.5 Sample Quality and Additional Benchmarks

Figure 12 shows random samples from the MNIST digit ablation study. Additionally, we also tested our architecture on the FashionMNIST dataset. Results are depicted in table 1

| Benchmark | | |
|---|---|---|
| Dataset | Base Distribution | NLL |
| FashionMNIST | Normal | -1264.539 |
| | Laplace | -1341.916 |
| | $R_{\text{logN}(1, .5), 1, 784}$ | **-1386.659** |

**Table 1:** Additional Benchmarks comparing VeriFlow with various base distributions. For each trial, a model with 10 alternations of LU- and additive coupling layers has been trained. Each coupling layers consists of a conditioner with 3 hidden layers. Each layer consists of 300 neurons. The same setup was used for the MNIST benchmark.

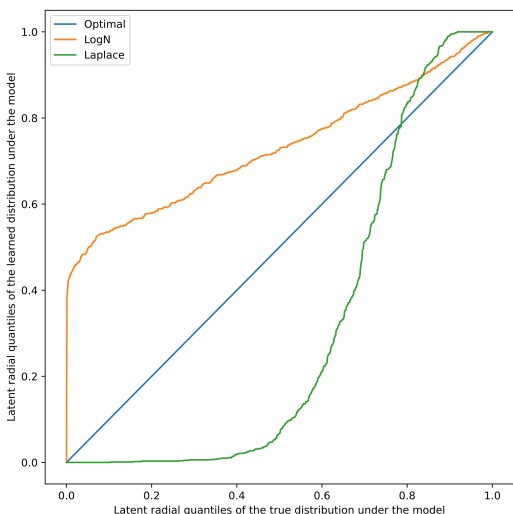

**Fig. 11:** $QQ$-plot of the empirical latent 1-norm distribution under the flow, $p_{\left|F^{-1}(X)\right|_1}$, against the theoretical 1-norm distribution induced by the base distribution. With both base distributions, the model struggles to match the empirical with the optimal latent 1-norm distribution in the low quantiles (although in opposite directions). However, with the log-normal distribution, the $QQ$-plot indicates approximately proportional tail behavior, while the empirical distribution is too wide in both directions with a Laplacian base distribution.

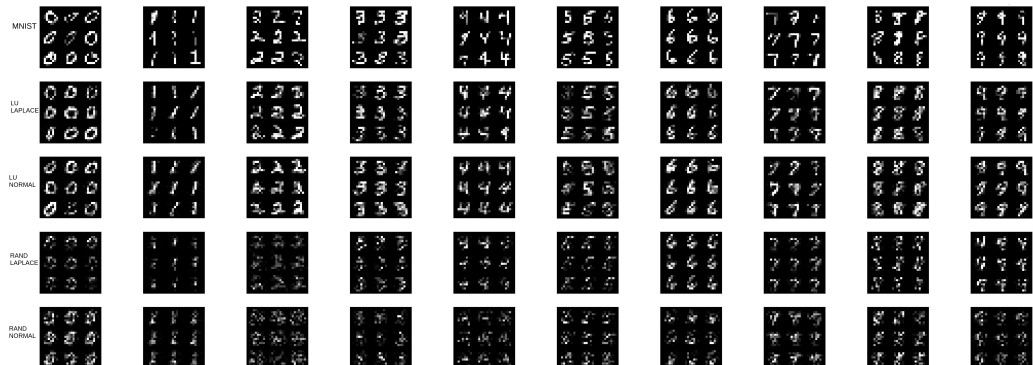

**Fig. 12:** Random samples from the ablation study on the MNIST digits. The $i$th column shows samples of the flow architectures trained on $\mathrm{MNIST}_i^{10\times10}$. Each row shows a different architecture (Top to bottom: $\mathrm{MNIST}_i^{10\times10}$ ground truth, LU + Laplace distribution, LU + Normal distribution, Random mask + Laplace distribution, Random mask + Normal distribution).

