# OpenReview forum: "VeriFlow: Modeling Distributions for Neural Network Verification"
_ICLR.cc/2025/Conference — Submitted to ICLR 2025_

### Official Review · Reviewer_UgWL · 2024-10-25

**Soundness:** 3
**Presentation:** 2
**Contribution:** 3
**Rating:** 5
**Confidence:** 3

**Summary:**

The paper introduces a novel framework to extract meaningful counterexamples that a neural network missclassifies called VeriFlow.
This is realized by restricting the global input domain only to images that probabilistically match the actual data distribution via transforming an assumed base distribution from a latent space to the data space.
The (non-)existance of counterexamples is then formally verified with existing NN verifier (Marabou/ERAN) on the generated data spaces,
where the input set for the verification query is determined by the desired confidence level on the base distribution.
The experiments show promising results and provide qualitative counterexamples.

**Strengths:**

- Providing meaningful counterexamples is highly relevant in the safety domain, which is not necessarily the case if they are generated form the entire input domain.
- The underlying transformations are piece-wise linear and the input set in the latent space corresponds to an L_p-ball, such that existing NN verifiers can be used to formally show the (non-)existance of counterexamples.
- The quality of the counterexamples can be selected by choosing a confidence level of the underlying distribution (e.g., clearly visible in Appendix C.3).
- Modelling the approach with distributions that go beyond naive ones and perform rigorous theoretical analysis on them.

**Weaknesses:**

Major points:

One common limitation of probabilistic approaches is that they assume a certain distribution and (in this case) transform this distribution via learning (!) to obtain desired properties (in this case, that it models the actual distribution of the data, e.g., outputs the set of "8" figures). However, if the learned (transformed) distribution in the data space actually matches the training set distribution is not adequately shown/tested.
For example, it is not said how long this has to be trained for until this property is reached (e.g., the loss is modeled in a way such that it becomes 0 once all desired properties are (formally) fulfilled). If this cannot be done, one could at least test it empirically: E.g., line 358 states that choosing a certain threshold obtains us top 1% typical images. Thus, as the paper can enclose the respective set in the data space, one could test if 1% of the training data is contained in that set and qualitatively assess if these could be classified as "typical". Doing so for different thresholds and classes would greatly strengthen that claim. If one cannot support the claim of modeling the actual data distribution in any way, there are no probabilistic guarantees even though distributions are used and I would rather categorize it as a heuristic.

The related work section does not adequately place this paper in the broader context of literature. Additionally, I found that both Sec. 2 and Sec. 5 contain stuff that are usually found in related work section and limitations/future work mentioned in Sec. 5 should be stated more clearly in a different section, e.g., Sec. 6. I propose to rework these sections: For example, a new Sec. 2 could give more background information on modeling distributions to get a better understanding of the subsequent Sec. 3.
The new related work section should place this work into the broader context:
- There are many more verifiers than presented in the paper. A good starting point for that literature research would be the verifiers of VNN-COMP. Also, not all optimization-based approaches are complete (i.e., either verify or provide a counterexample) but can also be incomplete and sound (i.e., if they say the property holds, it actually holds but there might be cases where they just return unknown without providing a counterexample; as is described for abstract interpretation). I think it would be better to state this, cite a broader range of NN verifiers, but you can be less detailed as currently in Sec. 2 as this is not the main focus of the paper.
- Sec. 5 also mentions "preprocessing steps" to ease verification. One should also mention adversarial training methods that incorporate NN verification into the training process, e.g.
[1] Gowal, et al., “Scalable verified training for provably robust image classification,” 2019.
[4] Zhang et al., “Theoretically principled trade-off between robustness and accuracy,” 2019.
[3] Muller, et al., “Certified training: ¨ Small boxes are all you need,” 2023.
[4] Koller et al., “End-to-end set-based training for neural network verification, 2024

Merging these sections as suggested might also leave more space for explaining the theoretical background in a new Sec. 2 on the required distributions, transformations, and terms like "constant Jacobian determinant" for the sake of self-containment. E.g. line 246 says that the constant Jacobian determinant is demanded, but it is not said how this is done.
Some propositions are also missing proofs (e.g., Prop. 1). If they are known results, they could be moved to the new Sec. 2 and cited adequately.

Minor points:
- The paper states that counterexamples generated from the global domain result in noisy, synthetic data that do not necessarily correspond to actual images. However, it is often not difficult to find counterexamples in the local neighborhood of images, which are still meaningful.
- Line 325 states that one has to learn a different transformation for each class. This could be stated more clearly.
- As the input set in the latent space corresponds to an L_p ball (a continuous connected set) and the applied transformations are continuous (piece-wise linear), also the set in the data space is continuous and connected. Thus, we implicitly assume that all our data corresponding to one class live in one connected space. This limitation should be better addressed by saying in which applications this assumption holds.
- Line 195: P_D not introduced, only probability density function p_D. Is it the respective CDF?
- Lack of intuition why a certain base distribution is chosen other than "it turned out that it boosted the performance"
- More detailed steps / motivation / intuition in Sec. 3 to make it easier for the reader to follow along. For example, one could (re-)state the desired properties of the architecture in Sec. 3.1 before saying how the components are constructed to better motivate why they are constructed like that.
- Introduce variables before they are used: e.g., line 349: What is y? Is it scores as introduced in line 356? But there, y is written in bold font and not in regular font as in line 349.
- Line 391: Which "input space" is increased? The dimension of the latent space? If not, how large is the latent space?
- Spelling/Grammar mistakes: Line 198: "Gaussian is ~the~ by far"


Other points that did not directly influence the score
- Rectified Linear Units are commonly abbreviated by ReLU, not ReLu.
- Clean figures (especially in appendix): You used titles of each image to display information. However, if something relates to an entire row / column, it's usually easier to read if you only write it as ylabel of the respective left most image / title of respective top image, respectively.

**Questions:**

- I want to know more about the enclosed set in the data space: How restrictive is the subset of the entire domain? For example, the network g_8 aims to only output images with an 8 displayed. However, are there 8s (from the training set) that cannot be constructed? Are a certain % of the 8s from the training set included in the exact / outer-approximative set in the data space if the input set in the latent space matches the same % threshold q?
- How would you address the limitation of the continuous connected set assumption on the data of each class?
- Regarding scalability of NN verifiers: Marabou regularly participates in VNN-COMP where much larger networks and also arbitrary computation graphs are considered. What specifically (e.g., types of layers) limits the scalability of your approach here? Line 390 also says that the verificaiton was done within a few seconds, which does not sound like a huge limitation.
- Many verifiers can also efficiently handle other activation functions than ReLU. Why is this limitation necessary?

---

> ### Author Response · Authors · 2024-11-25
>
> Thank you for the constructive feedback and providing helpful additional references.
>
> Regarding the empirical evaluation of the quality of the flow model, we would like to point out that we did the proposed analysis in the supplementary material C.4. As our analysis shows, it is possible to train models that encompass the entire dataset within the center of the learned distribution (although we still observed discrepancies in the lower quantiles), see also Figure 11.
>
> Minor point 1: If the neural network is buggy with regards to the global property already on a local level, then finding counterexamples in the nearby neighborhood of images may indeed be simpler and meaningful. The premise of this paper, however, is verifying global properties without such additional assumptions.
>
> Point 3:
> We want to provide some more context first. Our flow model does not transform a manifold or a set. Rather, it transforms the whole latent space to the target space bijectively. Therefore, there is no restriction that the data needs to live in a connected space.
> With that said, in the verification experiments of our paper, we indeed restrict the input space because we are interested in the most typical representatives of the class. In this case, the continuous transformation also implies that the transformed set is connected, although arbitrarily complex. We want to emphasize that this is solely due to the semantics of the property we verify in our experiments. In other words, we can increase the verified input space arbitrarily large and also verify a property over multiple (disconnected) sets.
> Figure 10 compares the quality of the samples from the unrestricted space and from the UDL as an additional insight.
> On a side note we want to mention that there is in fact a different restriction that can be derived from the nature of the transformation. That is, transforming a fully supported distribution implies that the resulting distribution will also be fully supported. However, it is possible to approximate any distribution arbitrarily well with a fully supported distribution. Indeed, this is a general restriction of normalizing flows and it is well known that many normalizing flow architectures have the universal approximation property.
>
> Point 4: $P_D$ denotes the probability that is induced by $p_D$, i.e. $P(x\in S) = \int_S p_D(x)dx$
>
> Point 5: Please note that the choice of 1-radial distributions is motivated by the fact that this allows us to describe density level sets via linear constraints in the latent space. Additionally, Appendix C.4 gives additional intuition on the choice of the norm-distribution. Together this gives a complete picture regarding the choice of the base distribution.
>
> Minor point 7:
> The variable y is introduced in Line 356 after it was used already, we will fix it.The variable y is a vector of length 10 that corresponds to the logits of the classifier. Both the bolded and unbolded y are the same variable and were printed differently due to a formatting error.
>
> Minor point 8, the bounds for each dimension of the latent space were increased. Note that the bounds for each dimension are kept very narrow  since the full positive verification (and not only finding a counterexample) with Marabou has a high runtime as soon as the input space exceeds a certain threshold shown in the plot 4b. The small space corresponds to a small percentage of the whole input space (around 1e^-7%).
>
> Question 3, we don’t see a major difference in network sizes compared to the networks used in VNNComp24. For larger networks, the verification with Marabou also usually times out according to VNNComp24 results. [5,6]
> An explanation for the rather minor performance differences in performance of Marabou in VNNComp24 compared to our instance may be the use of different Sofware (Gurobi vs built-in backend) and Hardware (AWS m5.16xlarge vs i7 CPU on a laptop). [7]
>
> Question 4: It is true that many verifiers, especially those that rely on interval propagation methods can handle sigmoid activation functions. Even Marabou 2 is able to handle Sigmoid layers. However, these methods typically rely on an initial overapproximation and an iterative refinement for these non-linearities. Verification of neural networks that contain sigmoid layers can lead to “Unknown” verification results. In contrast, ReLU layers can be precisely modeled in MILP formulations and allow for complete verification. [4]
>
> Thank you also for the other feedback points that we did not address explicitly. We will incorporate them into our paper.
>
> [4] https://dl.acm.org/doi/abs/10.1145/3551349.3556907
>
> [5] https://github.com/ChristopherBrix/vnncomp2024_results/blob/main/marabou/2024_cifar100/results.csv
>
> [6] https://github.com/ChristopherBrix/vnncomp2024_results/blob/main/marabou/2024_cgan_2023/results.csv
>
> [7] Slide 31: https://docs.google.com/presentation/d/1RvZWeAdTfRC3bNtCqt84O6IIPoJBnF4jnsEvhTTxsPE/edit#slide=id.g13dbba32db0_0_219

---

> > ### Comment · Reviewer_UgWL · 2024-11-25
> >
> > Thank you for your clarifications. I increased my score as many of my points were somewhat addressed. The overall approach is promising. but I still think there is some work to do, as discussed in my initial review. Best of luck with your submission!

---

> > > ### Author Response · Authors · 2024-11-25
> > >
> > > We noticed that we have not explicitly mentioned in this rebuttal that we added a new Section D in the updated PDF, which now contains experimental results on full MNIST to further address the scalability concerns. We will incorporate these experiments into the main body of the paper. This comment is provided in case changes to the PDF are not automatically shown to the reviewer.
> > >
> > > Thank you again for the helpful feedback and we appreciate that you find the overall approach promising.

---

> > > ### Author Response · Authors · 2024-12-02
> > >
> > > We updated the paper and want to summarize the changes relevant to this discussion:
> > >
> > > * We enhanced our related work section to include all the references you explicitly listed and additional references to put our work better into context. In particular, we outlined the difference between complete and incomplete verifiers
> > > * We included experimental results with the higher dimensional dataset MNIST 28x28 into the main body of the paper. Starting from Line 395.
> > > * We are now more concise in the background section.
> > > * We added a paragraph on the organization of the paper before Section 3.
> > > * We stated more clearly that in our experiments that we trained a flow model for each class individually
> > > * We moved the paragraph discussing the limitations from Section 5 to Section 6
> > > * We fixed errors in notation, grammar and figures. In particular the ones you listed in the review.
> > >
> > > We thank you again for your feedback that helped improve the paper.

---

> > > > ### Comment · Reviewer_UgWL · 2024-12-03
> > > >
> > > > I very much appreciate your additions and I already updated my score from my initial rating because of it.
> > > >
> > > > However, after re-reading the paper (especially Sec. 3), my first major point about the properties of a learned (!) transformation was not addressed. In fact, the final Prop. 4 only states that the designed network is a "uniformly scaling piecewise affine flow". This property seems to hold independently of how long you train the network, so it also holds for a randomly initialized network. As such a network cannot reflect the data distribution, the questions raised in my first major point remain.

---

> > > > > ### Author Response · Authors · 2024-12-04
> > > > >
> > > > > We thank you again for the constructive discussions and we are happy to provide more context on your first major point.
> > > > >
> > > > > It is true that our verification approach cannot provide probabilistic guarantees if the underlying data distribution is not captured appropriately. Our approach inherits this limitation from the neuro-sybolic verification framework [1]. However, various potential mitigation strategies have been identified. For example, regulatory institutions can issue well trained specification networks such as the flow model along with the properties. Alternatively, public challenges and community efforts are conceivable. More generally, our approach of using a (possibly unsafe) neural network as part of the specification is not a novel idea [1,7,8] and is also justified by the fact that formalizing system requirements that go beyond local robustness is notoriously difficult and error-prone also for humans [2, 4, 5, 6].
> > > > >
> > > > > Beside these conceptual considerations, we have put quality assurance measures in place that help mitigate this risk of a misleading verification.
> > > > >
> > > > > On a theoretical level, we would like to point out that training with the maximum likelihood objective, as we did, is equivalent to minimizing the KL-divergence between the learned distribution and the data distribution. In particular, minimizing the KL-divergence corresponds to the property that our flow model captures the true distribution. More precisely, the KL-divergence between two distributions is 0 iff the two distributions coincide. If the computational resources, the number of samples and the capacity of the hypothesis space go to infinity, the solutions will converge to the true distribution. However, note that the likelihood objective does not directly allow to infer the exact value for the KL-divergence. In practical scenarios one therefore needs to use common convergence tests to obtain a stopping criterion. This is also the strategy we followed in our experiments.
> > > > >
> > > > > On a practical level, we analyzed our models in Appendix C.4 and compared the defined and the empirical norm-distribution via Q-Q-plot in Figure 11 as you suggested in your initial comment. It effectively plots the UDL percentage against the percentage of the test data that is contained in the UDL. Our results demonstrate that it is possible to train models that encompass the entire dataset within the center of the learned distribution (although we still observed discrepancies in the lower quantiles). In addition, we also explored custom base distributions in order to improve the calibration of our models (Fig. 8). We agree that it is important to add a brief summary of these experiments and discussion in the main body of the paper.
> > > > >
> > > > >
> > > > > References:
> > > > >
> > > > > [1] Xuan Xie et al. “Neuro-Symbolic Verification of Deep Neural Networks”. In Proceedings of the Thirty-First International Joint Conference on Artificial Intelligence, (IJCAI) 2022
> > > > >
> > > > > [2] Jonathan P. Bowen. “Gerard O’Regan: Concise Guide to Formal Methods: Theory, Fundamentals and Industry Applications”
> > > > >
> > > > > [3] Pierre-Jacques Courtois et al. “Licensing of safety critical software for nuclear reactors. Common position of international nuclear regulators and authorised technical support organisations.”
> > > > >
> > > > > [4] Antti Pakonen et al. “User-friendly formal specification languages - conclusions drawn from industrial experience on model checking”
> > > > >
> > > > > [5] Kristin Yvonne Rozier. “Specification: The Biggest Bottleneck in Formal Methods and Autonomy”
> > > > >
> > > > > [6] Rainer Schlör et al. “Using a Visual Formalism for Design Verification in Industrial Environments”.
> > > > >
> > > > > [7] Toledo, F. et al. “Distribution models for falsification and verification of dnns”.
> > > > >
> > > > > [8] Wu, H. et al. “Toward certified robustness against real-world distribution shifts.”

---

### Official Review · Reviewer_d2FL · 2024-11-02

**Soundness:** 4
**Presentation:** 2
**Contribution:** 4
**Rating:** 6
**Confidence:** 4

**Summary:**

This paper proposes a novel approach to modeling input distributions for the purpose of formal verification of neural networks. In particular, the authors leverage flow models to transform input space into upper density level sets, by which existing formal verification approaches for deep neural networks are restricted to search in those data distributions of interest. In this way, one can find more meaningful counterexamples that have practical meaning or are computable under certain perturbation approaches. The authors identify two main properties that shall be satisfied by flow models suited to formal verifications. One is that the transformation by flow models must be peicewise affine. The other is that UDL sets takes the shape of L^p-ball in the latent space. Based on the two properties, the authors further analyze several existing flow models that meet the requirements and demonstrate the effectiveness when being applied to the verification of DNNs with SOTA tools such as ERAN and Marabou 2.0. The authors evaluate their proposed approach and demonstrate the effectiveness in finding more meaningful counterexamples (or adversarial examples) and the scalability to different types of verification approaches.

**Strengths:**

1. A novel perspective to neural network verification. Most of the existing approaches try to define tightest-possible over-approximation for DNNs to be verified to reduce over-estimation and false positives in formal verification results. This paper considers the verification from a new perspective by restricting verification approaches to meaningful input space. In literature, there are several attempts to the formal verifications of semantic perturbations such as rotations, occlusions, and geometric transformations. This paper considers the semantic perturbations from the distribution perspective. This imposes a new verification problem that is different from existing formal verification problems of DNNs. The work would inspire more solutions to the new problem.

2. The proposed flow-based method is technically sound and practically useful. Transforming original input space into UDL sets by piecewise affine and abstracting the sets using effectively computable abstract domains are intuitively applicable and straightforward, However, the difficulty lies in finding appropriate flow models and abstractions. The authors identifies the conditions and theoretically prove the qualification of identified flow models to the verification task.


3. The evaluations are comprehensive and the results are convincing. The results show that the counterexamples computed in the proposed way are indeed more interpretable, while it is compatible with mainstream verification approaches such as SMT-based or abstraction-based ones. The overhead caused by flow models can be ignored. The verification efficiency is mainly depended on the backend DNN verification approaches. However, this is applicable only to low-dimensioned cases.

**Weaknesses:**

1. The presentation shall be carefully proofread if the paper is finally accepted. There are a lot of grammatical errors that could be completely avoided. For instance, on Line 212, the sentence let XXX is the hyper volume, and on Line 480 complext problems should be complex problems. I just name a few of them. The paper seems to be written in haste. That makes me lower my score.

2. The experiment part should provide more evidences about meaningful counterexamples generated by the proposed approach. As motivated by finding more meaning counterexamples from neural network verification, the authors should give more evidences. They shall not be placed in appendix.

3. Section 2 (verification background) seems not necessary. The propose modeling methods are not closely related to backend verification approaches. Having section 2 or not will not affect the understanding of the paper. It provides few useful information to help readers understand the paper.

**Questions:**

1. In Definition 2, X is a called $k$-radially distributed. However, there is no assumption on $k$. Where does $k$ come from?

2. It is wired to have only one subsection Section 3.1 inside Section 3. Besides, why do you consider only the types of the first three types in F? In which principle one can decide the types for the first three layers and other remainder layers for different distributions?

3. In the paper, it is not clear how the flow model is trained and how the quality of trained flow models affect the verification results. Can you provide more details regarding the the issues?

---

> ### Author Response · Authors · 2024-11-25
>
> We thank you for your detailed and constructive feedback.
>
> Regarding the weaknesses, we will try to be more concise in the background section and include experiments with the unscaled original MNIST from the new Section D into the main corpus of the paper.
>
> Next, we want to address the questions:
>
> Q1: Thanks for pointing this out. $k\in \mathbb{N}_{>0}\cup\{\infty\}$
>
> Q2: Thanks for the suggestion. We will consider restructuring Section 3. Regarding the question on the types of layers: We assume that you refer to the restriction given in Proposition 4. Here it is just the case that other layer types do not need any additional restrictions to guarantee that the transformation will be piecewise affine and uniformly scaling. There is no fully principled guide on how to choose the layers of a normalizing flow according to the distribution. There are a few heuristics that can be mentioned: As we argued in the paper, alternating LU layers and additive coupling can have positive effects. Additionally, masked additive convolutions can be a good choice for high resolution image data. When working only with additive coupling layers, as in the original NICE architecture, it is advisable to put a permutation layer before each coupling layer and to append scaling as the final layer since otherwise the determinant is constrained to be $1$.
>
> Regarding Question 3, the flow is trained by the standard maximum likelihood objective, which minimizes the KL-divergence between the learned and the true distribution. The objective guarantees that in the infinite data and capacity regime the solution will converge to the true distribution.
> In general, an insufficiently trained flow model can contain out-of-distribution data in the upper density level sets and may also assign low densities to images from the training data, compared to a well-fitting flow model. Therefore, a poorly trained flow model will lead to poor verification results. More precisely, positive verification certificates may not actually guarantee the property to hold within the data distribution and counterexamples returned by the verification tool for when the property does not hold may be an out-of-distribution data point (e.g., an image unrecognizable as a digit).

---

> > ### Comment · Reviewer_d2FL · 2024-11-26
> >
> > Thanks very much for the clarifications. The motivation of the paper becomes clearer and the approach is novel, compared with other existing verification approaches. The authors discussed the scalability of their approach. But in the paper, it appears like it depends on the scalability of backend verifiers. How about the performance of the approach when it is applied to other more complex tasks such as CIFAR?

---

> > > ### Author Response · Authors · 2024-11-26
> > >
> > > Thank you for your feedback and the question.
> > >
> > > To be more clear regarding the scalability, we want to emphasize the difference between creating specifications and verifying those specifications. Creating specifications boils down to training an adequate flow model. This step is based on gradient descent and therefore is also possible for higher dimensional data sets such as MNIST 28x28.
> > > However, the verification of the generated specifications are subject to the scalability of the verification tool at hand.
> > > In our experiments, we aim to illustrate the differences in scalability between ERAN and Marabou for verifying these specifications. That is the reason why we use both of them on MNIST 10x10.
> > > We will make this interconnection clearer in the paper as well.
> > > Also note that in the new Section D, we use only the more scalable verifier ERAN for the verification of the larger network as Marabou times out after one hour.
> > >
> > > Regarding the CIFAR dataset, we could not yet obtain a well-trained flow model for now.
> > > More precisely, we trained models with the same components as used with MNIST 28x28 on the CIFAR dataset but observed that the model contained out-of-distribution data in the upper density level set.
> > > We believe this limitation can be tackled by using additive convolution layers and a multiscale architecture. These components are already implemented but were not yet used into our experiments.
> > >
> > > Thank you again for your feedback.

---

> > > ### Author Response · Authors · 2024-12-02
> > >
> > > We updated the paper and want to summarize the changes that are relevant to this discussion:
> > >
> > > * We included experimental results with the higher dimensional dataset MNIST 28x28 into the main body of the paper. Starting from Line 395.
> > > * We are now more concise in the background section.
> > > * We enhanced our related work section.
> > > * We added a paragraph on the organization of the paper before Section 3.
> > > * We fixed errors in notation, grammar and structure, in particular the ones you listed in the review.
> > >
> > > We thank you again for your feedback that helped improve the paper.

---

### Official Review · Reviewer_rKgE · 2024-11-03

**Soundness:** 2
**Presentation:** 1
**Contribution:** 2
**Rating:** 5
**Confidence:** 4

**Summary:**

**Outline**

The authors propose VeriFlow a normalizing flow-based method to learn more realistic input specifications for Deep Neural Network (DNN) verification which can capture more diverse inputs than existing local $L_{p}$ norm-based robustness specification while excluding noisy meaningless inputs included in global input specifications. The authors show with the help of the proposed method they can generate more sensible counter-examples that violate the property under verification.

**Strengths:**

**Strengths**

- In theory, I think coming up with sensible specifications for Neural Networks is an important problem. Unfortunately, there does not seem to be enough work in this direction.

- Using normalizing flows seems to be a reasonable choice for encoding input specification.

**Weaknesses:**

**Questions and weaknesses**

I had a really hard time understanding the paper. The organization and notations and definition used in the paper are not mentioned beforehand.

**Motivation of the work**\
Q1. (Lines 52 - 53) “Local properties, on the other hand, suffer from the same problem as statistical testing, i.e., they rely on a high-quality data set that the verification property is based on.” - I completely agree with this statement. However, even networks trained with certifiably robust methods on datasets like CIFAR-10 and Tiny ImageNet [1] (excluding MNIST) still have quite low verified accuracy (percentage of verified local properties), even for small epsilon values like $\epsilon = 8/255$. Given that networks struggle to achieve robustness on this relatively "high-quality data," as the authors noted, what is the reasoning for moving to more challenging input specifications?

**Doubts regarding Proposition 2**

Q1. I find proposition 2 hard to understand possibly due to the error/ambiguity notations and undefined terms. I list my concerns below \
a. What are affine regions $R_{1},\dots, R_{n}$? \
b. Affine regions and their relation with $F: \mathbb{R}^{n} \to \mathbb{R}^{n}$ are not defined formally. \
c. Why is the number of affine regions $R_{1},\dots, R_{n}$ the same as the input dimension of F? In the worst case, what is the number of affine regions? Is it exponential w.r.t number of activations? \
d. What is $d$ in $\mathbb{R}^{d}$ ?

Q2. Also, piecewise affine functions are not differentiable (like $ y = ReLU\(x\) $ is not differentiable ). Can you give an example of how proposition 2 holds for this case i.e. $y = ReLU(x)$?


**Usefulness**

Q1. I am doubtful about the usefulness of the proposed method and seems only experiments are with downsized (10x10) MNIST datasets. Can authors provide any insights into how the proposed method can generate preconditions to other scenarios MNIST (28x28), CIFAR10 (3x32x32), etc.?  It is well known that SMT-based solvers and verification w.r.t. global properties do not work with high-dimensional data, does that mean the proposed can only work for low-dimensional inputs?


**Representation & writing**

Q1. The authors should have clarified which specification the provided counterexample was violating. Let me assume it was a global property. If so, does not this counterexample substantiate why most well-known works [2, 3] focus on local properties? When violations occur, at least local properties tend to produce more sensible, human-understandable images.

Q2. (Lines ) “One well-studied example for a local property is adversarial robustness which requires the neural network to classify any point from the data set as the same class as any minor perturbation of that point.” - Correct me if I am wrong but I don’t think this is an accurate definition. Suppose an image of a 5, along with all its minor perturbations, gets classified as a 6. In this case, the predicted class remains consistent, but all labels are incorrect. Typically, in adversarial robustness, this scenario is considered a violation of the output specification. According to the authors’ definition, however, it would not be regarded as a violation.

Q3. The presentation of the paper could be clearer. Specifically in Section 3, I am unsure which contributions are original to this paper and which pre-exist in the normalizing flows literature.

[1] “Connecting Certified and Adversarial Training”, NeurIPS, 2023.
[2] “General Cutting Planes for Bound-Propagation-Based Neural Network Verification”, NeurIPS, 2022.
[3] “Complete Verification via Multi-Neuron Relaxation Guided Branch-and-Bound”, ICLR, 2022.


**Related work is obsolete**

The works cited in this paper are outdated and have been surpassed by more recent research. I strongly recommend that the authors include references to papers from leading ML conferences (such as ICML, NeurIPS, and ICLR) and top programming languages/verification venues (such as POPL, PLDI, and CAV) from the past 3-4 years. For example,

**Abstract Domain:** The Zonotope abstract domain has been surpassed by the DeepPoly domain [1]. Subsequently, multi-neuron abstraction was introduced in [2], and more recently, the DiffPoly domain [3] was proposed for hyperproperty verification.

**Branch & Bound-Based Verifiers:** Current state-of-the-art verifiers for local properties, such as GCP-CROWN [4] and MN-BaB [5], are not mentioned.

**Input Specification:** Unlike this work, many existing studies consider weaker local specifications to model practical attack scenarios, such as robustness against geometric perturbations [1, 6] and robustness against universal adversarial perturbations (UAP) [3, 7, 8]. This work should mention these studies and justify its approach of addressing more challenging input specifications.

[1] “An Abstract Domain for Certifying Neural Networks”, POPL, 2019.\
[2] “PRIMA: General and Precise Neural Network Certification via Scalable Convex Hull Approximations”, POPL, 2022.\
[3] “Input-Relational Verification of Deep Neural Networks”, PLDI, 2024.\
[4] “General Cutting Planes for Bound-Propagation-Based Neural Network Verification”, NeurIPS, 2022.\
[5] “Complete Verification via Multi-Neuron Relaxation Guided Branch-and-Bound”, ICLR, 2022.\
[6] “Certifying Geometric Robustness of Neural Networks”, NeurIPS, 2019.\
[7] “Towards Robustness Certification Against Universal Perturbations”, ICLR, 2023.\
[8] “Relational DNN Verification With Cross Executional Bound Refinement”, ICML, 2024.

**Questions:**

Refer to the Weaknesses section

---

> ### Author Response · Authors · 2024-11-25
>
> Thank you for the detailed and constructive review.
>
> Regarding the Motivation of the work, it is true that local properties such as local robustness are hard to ensure, even when training with specialized robust training methods.
> Indeed, since global properties are subject to the whole input space, the problem of ensuring global properties in a neural network is arguably harder than ensuring local properties that are restricted to a ball around each data point.
> However, the need for ensuring global properties such as global fairness constraints exists independent of our work. [1]
> Furthermore, our flow model actually relaxes the global property because using the flow model to restrict the input space to the “center” of the data distribution removes the necessity of the neural network to ensure the global property on the whole input space, including noise.
> In other words, our work aims to push the challenging verification of global properties more towards local properties by restricting the input space to the data distribution.
>
> Regarding your questions in context of Proposition 2, we first apologize that there was a typo that mixed up the number of affine regions n and the dimension of the space d. We first clarify the term affine region. By definition, a function $f: X\to Y$ is piecewise affine, if there is a partition of the domain $X = X_1 \cup \cdots \cup X_n$ such that $f$ restricted to $X_i$ is affine for all i. We call $X_1,\ldots, X_n$ affine regions of $f$.
> Proposition 2 states that the change of variables formula is valid for piecewise affine functions, if the affine regions are Borel sets and the determinant is computed piecewise.
>
> Note that the proposition is not applicable to ReLU because ReLU is not a bijection. However, leaky-ReLU is a good example to understand the statement.
> In this case, the density of $Y = \text{leaky-ReLU}_{\alpha}(X)$ can be computed as follows:
> - $p_Y(y) = p_X(\text{leaky-ReLU}_{\alpha^{-1}}(y)) \cdot 1 ~~~~~~~$ if $y\geq0$
> - $p_Y(y) = p_X(\text{leaky-ReLU}_{\alpha^{-1}}(y))\cdot \alpha^{-1}~~~$ if $y<0$
>
>  where we use that $\text{leaky-ReL}U_\alpha^{-1} = \text{leaky-ReL}U_{\alpha^{-1}}$.
>
>
> Regarding the usefulness/scalability of our approach, we want to emphasize the difference between creating specifications and verifying those specifications. Creating specifications boils down to training an adequate flow model. This step is also possible for higher dimensional data sets. To show this, we added Section D in the paper showing samples of a trained model on MNIST 28x28.
> However, the verification of the generated specifications are subject to the scalability of the verification tool at hand. In that case, Marabou fails to verify the property within a timeout of one hour in our experiments.
> More scalable verifiers like ERAN, however, can perform the bound propagation also on such larger networks. We also included plots for verification using ERAN in the paper in Section D. We will incorporate these experiments in the final version of our paper.
>
>
> Regarding Question 1 in Section representation & writing: In all our verification experiments we verify global properties. The specification that was violated is described textually in the Introduction in Lines 65-66 and more formally in Lines 420-423.
> We see verification of local properties and global properties as complementary challenges. It is correct that local properties naturally produce human readable counterexamples, and we aim to achieve the same for global properties with our work.
>
> Regarding Question 2, popular definitions do not explicitly state the necessity for the classification on the original point to be correct. An excerpt from [2]:
> “It is often required that the original input x is classified correctly, but this requirement can vary
> across papers. Some papers consider x′ an adversarial example as long as it is classified differently from x.”
>
> Q3: We acknowledge that the original contributions presented in Section 3 should be better separated from the standard definitions regarding normalizing flows. We would like to mention that to the best of our knowledge, the propositions in Section 3 have not been previously stated in the normalizing flow literature.
>
> We thank you for the additional references that we will incorporate into our related work section.
>
> References:
>
> [1] “Verifying Individual Fairness in Machine Learning Models”, UAI 2020. http://proceedings.mlr.press/v124/george-john20a/george-john20a.pdf
>
> [2] “On Evaluating Adversarial Robustness” Footnote on Page 5: https://arxiv.org/pdf/1902.06705

---

> ### Comment · Reviewer_rKgE · 2024-11-26
>
> Thank you to the authors for their detailed response. Overall, I am satisfied with the answers and am willing to increase the score, provided the authors update the PDF to reflect the promised changes, specifically by including a corrected version of Proposition 2 and an updated discussion in the related work section. Additionally, I would appreciate clarification on the following points:
>
> - In the worst case, what is the number of affine regions? Is it exponential with respect to the number of activations?
> - Does this pose a bottleneck for scaling the proposed technique to high-dimensional data?

---

> > ### Author Response · Authors · 2024-12-02
> >
> > Thank you again for your helpful feedback. We updated the paper and want to summarize the changes relevant to this discussion:
> >
> > * We enhanced our related work section to include all the references you explicitly listed and additional references to put our work better into the context of recent publications.
> > * We corrected Proposition 2 and provided intuition for it.
> > * We included experimental results with the higher dimensional dataset MNIST 28x28 into the main body of the paper. Starting from Line 395.
> > * We added a paragraph on the organization of the paper before Section 3.
> > * We are now more concise in the background section.
> > * We fixed errors in notation, grammar and structure.
> >
> >
> > Regarding the questions:
> >
> > In the worst case, the number of affine regions indeed grows exponentially with respect to the number of activations.
> > However, other works have shown theoretically and empirically that the number of affine regions of ReLU networks are expected to be far below the exponential maximum for both random and trained weights [1,2].
> >
> > In addition, the runtime of training, sampling and density estimation is independent of the number of affine regions. This also holds for the computation of the determinant in Proposition 2. More precisely, the determinant of each additive coupling layer always has the constant value of 1 and these are the only layer types that contain ReLU nodes.
> > Overall, the expression from Proposition 2 is computed in linear time in the number of neurons, taking into account the various layer types such as LU-Layers for which the determinant can be computed in $O(d)$ due to their diagonal Jacobians.
> >
> > Regarding the scalability of the subsequent verification process, we can expect a higher runtime for verification, if the number of linear regions grows large. This is particularly true for SMT-Based verification tools. However, we want to emphasize, again, that this is only if the trained flow model that was used for verification had an unusually high number of affine regions than what can be expected according to the aforementioned findings [1,2].
> >
> > Additionally, in our experiments, we observed that the runtime for verification with our flow model was comparable with the runtimes during VNNComp24 when comparing to networks with similar number of parameters [3,4,5]. Minor differences may be due to our different hard- and software setup.
> > This indicates that our flow model architecture is not prone to having a higher number of linear regions compared to the reference networks used during the VNNComp24.
> >
> > We thank you again for your feedback that helped improve the paper.
> >
> >
> > References:
> >
> > [1] “Deep relu networks have surprisingly few activation patterns“ NeurIPS’19
> > https://proceedings.neurips.cc/paper_files/paper/2019/hash/9766527f2b5d3e95d4a733fcfb77bd7e-Abstract.html
> >
> > [2] “Complexity of Linear Regions in Deep Networks” ICML'19
> >     https://proceedings.mlr.press/v97/hanin19a.html
> >
> > [3] https://github.com/ChristopherBrix/vnncomp2024_results/blob/main/marabou/2024_cifar100/results.csv
> >
> > [4] https://github.com/ChristopherBrix/vnncomp2024_results/blob/main/marabou/2024_cgan_2023/results.csv
> >
> > [5] Slide 19 in https://docs.google.com/presentation/d/1RvZWeAdTfRC3bNtCqt84O6IIPoJBnF4jnsEvhTTxsPE/edit#slide=id.g279a2f344f5_1_0

---

> > > ### Comment · Reviewer_rKgE · 2024-12-03
> > >
> > > Thanks for your reply and the updated pdf. I am raising my score as promised earlier.

---

### Official Review · Reviewer_ekRG · 2024-11-05

**Soundness:** 3
**Presentation:** 4
**Contribution:** 4
**Rating:** 8
**Confidence:** 3

**Summary:**

This paper introduces a methodology to verify semantically meaning properties using a neuro-symbolic approach, in which the input to the neural network under verification is specified by a flow model trained to model a particular data distribution. The paper shows that the proposed method can be used to generate in-distribution counter-examples that violate a specified output property.

**Strengths:**

- This paper proposes a generic neuro-symbolic approach to limit the input space to a given data distribution. The specification considered here is novel and of practical interest.
- The paper designs a novel flow model that allows the definition of the pre-image of a density level set in the latent space via linear
constraints, making the model both interpretable and compatible with existing neural network verifiers.
- The paper instantiates the proposed verification method with two verifiers, one based on bound propagation, the other based on search, and illustrates the trade-off in performance and functionality.

**Weaknesses:**

- The paper only considers relatively small datasets. Validating the approach on common benchmark sets considered in VNN literature such as CIFAR-10 and ACAS-Xu would better illustrate the scalability of the approach.
- The soundness of the verification depends on the quality of the trained flow model. This is an intrinsic issue of neuro-symbolic approaches like this, but I do acknowledge that I cannot see an alternative approach to verify the specifications considered in the paper.

**Questions:**

How does the flow-based approach compare with the VAE-based approach for modeling a given distribution [1]?

[1] https://arxiv.org/pdf/2007.08450

---

> ### Author Response · Authors · 2024-11-25
>
> Thank you for your constructive feedback.
> We acknowledge the importance of validating our approach on larger benchmark datasets. We conducted additional experiments where we trained our flow model on full MNIST 28*28 to show its capability to train on higher dimensional datasets, too. We ran verification experiments only using ERAN and excluded verification with Marabou due to scalability issues that are inherent to the verifier. The results are now in the preliminary Section D in the paper and will be maintained in the main corpus in the final version.
>
> The specific paper in your question aims primarily at learning perturbation sets using a conditional VAE and thus, has a slightly different scope.
> In general though, while VAEs can be used to also obtain more typical counterexamples, they don’t provide the theoretic properties as our flow model such as bijectivity and mapping of high density level sets between the latent space and target space.
> However, the reference is still related and we will incorporate it in the related work section.

---

### Meta-Review · Area_Chair_PFpx · 2024-12-20

**Metareview:**

The submission proposes the VeriFlow architecture as a flow based density model tailored to allow any verification approach to restrict its search to the some data distribution of interest.

+ The topic is of interest as more meaningful specifications are important to showcase the usefulness of verification.
+ The proposed methodology appears to be technically sound.

- The paper is difficult to follow and requires a rewrite to improve its clarity.
- It is not clear how well the learned transformation represents the actual data distribution.

**Additional Comments On Reviewer Discussion:**

The submission has received mixed ratings. The reviewers carefully considered the author rebuttal and used the discussion period to seek further clarity on the submission. However, questions still remain about how well the proposed method captured the real data distribution of interest. Furthermore, the submission also lacks clarity. While the work has merit, it requires a major revision.

---

### Decision · Program_Chairs · 2025-01-22

Reject